# Recruitment of the CoREST transcription repressor complexes by Nerve Growth factor IB-like receptor (Nurr1/NR4A2) mediates silencing of HIV in microglial cells

**Fengchun Ye\***, **David Alvarez-Carbonell, Kien Nguyen, Konstantin Leskov, Yoelvis Garcia-Mesa, Sheetal Sreeram, Saba Valadkhan, Jonathan Karn**

Department of Molecular Biology and Microbiology, Case Western Reserve University, Cleveland, Ohio, United States of America

\* fxy63@case.edu

## Abstract

Human immune deficiency virus (HIV) infection in the brain leads to chronic neuroinflammation due to the production of pro-inflammatory cytokines, which in turn promotes HIV transcription in infected microglial cells. However, powerful counteracting silencing mechanisms in microglial cells result in the rapid shutdown of HIV expression after viral reactivation to limit neuronal damage. Here we investigated whether the Nerve Growth Factor IB-like nuclear receptor Nurr1 (NR4A2), which is a repressor of inflammation in the brain, acts directly to restrict HIV expression. HIV silencing following activation by TNF-α, or a variety of toll-like receptor (TLR) agonists, in both immortalized human microglial cells (hμglia) and induced pluripotent stem cells (iPSC)-derived human microglial cells (iMG) was enhanced by Nurr1 agonists. Similarly, overexpression of Nurr1 led to viral suppression, while conversely, knock down (KD) of endogenous Nurr1 blocked HIV silencing. The effect of Nurr1 on HIV silencing is direct: Nurr1 binds directly to the specific consensus binding sites in the U3 region of the HIV LTR and mutation of the Nurr1 DNA binding domain blocked its ability to suppress HIV-1 transcription. Chromatin immunoprecipitation (ChIP) assays also showed that after Nurr1 binding to the LTR, the CoREST/HDAC1/G9a/EZH2 transcription repressor complex is recruited to the HIV provirus. Finally, transcriptomic studies demonstrated that in addition to repressing HIV transcription, Nurr1 also downregulated numerous cellular genes involved in inflammation, cell cycle, and metabolism, further promoting HIV latency and microglial homoeostasis. Nurr1 therefore plays a pivotal role in modulating the cycles of proviral reactivation by potentiating the subsequent proviral transcriptional shutdown. These data highlight the therapeutic potential of Nurr1 agonists for inducing HIV silencing and microglial homeostasis and ultimately for the amelioration of the neuroinflammation associated with HIV-associated neurocognitive disorders (HAND).

**Data Availability Statement:** The RNA-seq data has been deposited in the NCBI Sequence Read Archive (SRA) and is available under BioProject

accession PRJNA789419. All other relevant data are within the manuscript and its Supporting Information files.

**Funding:** This study was supported by NIH grants R01 DA043159 and R01 DA049481 to J. K. and R21-AI127252 to S. V. and two Development Awards from CFAR P30-AI36219. The funders had no role in study design, data collection and analysis, decision to publish, or preparation of the manuscript.

## Author summary

HIV enters the brain almost immediately after infection where it infects perivascular macrophages, microglia and, to a much lesser extent, astrocytes. In previous work using an immortalized human microglial cell model, we observed that integrated HIV constantly underwent cycles of reactivation and subsequent re-silencing. We now show that HIV shutdown after proviral reactivation is mediated by the Nurr1 nuclear receptor. Both the functional activation of Nurr1 by specific agonists, and the over expression of Nurr1, resulted in rapid silencing of activated HIV in microglial cells. Nurr1 not only repressed HIV expression but also selectively down regulated genes involved in microglial homeostasis and inflammation. Thus, Nurr1 is pivotal for HIV silencing and repression of inflammation in the brain and is a promising therapeutic target for the treatment of HAND.

## Introduction

A large number of HIV infected patients develop HIV-associated neurocognitive disorders (HAND) [1]. Symptoms seen in well-suppressed people with HIV (PWH), range from the mild neurocognitive disorder (MND) to asymptomatic neurocognitive impairment (ANI) [2–4]. Although combination antiretroviral therapy (ART) dramatically lowers the levels of HIV RNA in the brain [5–7], it does not reduce the incidence of HAND [4,8]. Thus, with the availability of widespread ART, HIV-associated dementia is virtually eliminated, but the frequencies of MND and ANI have actually increased [4,8].

HIV invades the brain soon after primary infection [9]. The virus primarily infects perivascular macrophages and microglial cells, as well as a few astrocytes, but it does not infect neurons [10,11]. Because microglial cells are much longer-lived than astrocytes and perivascular macrophages, and can support productive HIV replication, they serve as long-lived cellular reservoirs of HIV-1, even in well-suppressed patients receiving ART [12–14].

It has been challenging to associate the direct effects of HIV infection of microglia with the development HAND. Initial studies indicated, paradoxically, that HAND did not correlate with the number of HIV-infected cells or viral antigens in the central nervous system (CNS) [15,16], but instead correlates strongly with systemic inflammation and CNS inflammation [17]. However, the early studies neglected the potential side effects of anti-HIV drugs on neuronal damage, which could mask the benefits of reduced HIV expression by ART and the impact of HIV latency which can obscure the true levels of HIV in the CNS.

A unique feature of HIV infection of microglial cells is that the virus is able to efficiently establish latency [18–21]. In microglial cells, transcription initiation is primarily regulated by NF-κB. In resting microglia, NF-κB is sequestered in the cytoplasm [18–20]. However, unlike memory T-cells, pTEFb is not disrupted [22,23]. The provirus is also silenced epigenetically through the CoREST and polycomb repressive complex 2 (PRC2) histone methyltransferase machinery [13,24–27]. Activation of microglia by pro-inflammatory signals, such as tumor necrosis factor-alpha (TNF-α) and interleukin-1 beta (IL-1β), reversed these molecular restrictions, inducing the viral production which could potentially lead to neuropathology.

The pro-inflammatory environment in the brain of HIV-infected patients is therefore likely to be highly permissive for HIV reactivation from latency [21]. However, microglia have developed counterbalancing mechanisms to prevent exaggerated activation. In the normal CNS environment, healthy neurons provide signals to microglia via secreted and membrane bound factors such as CX3CL1 and neurotransmitters that restore microglial homeostasis [28] and

induce HIV-silencing [19–21]. For example, using a co-culture of (iPSC)-derived human microglial cells (iMG) that were infected with HIV and neurons, we demonstrated that HIV expression in iMG was repressed when co-cultured with healthy neurons but induced when co-cultured with damaged neurons [21]. Secretion of HIV proteins such as transactivator of transcription (Tat), negative regulatory factor (Nef), envelope glycoprotein gp120, and viral RNA reinforce the activation cycle because they are directly neurotoxic and also contribute to inflammation in the brain by activating uninfected microglial cells [29–37].

It is therefore important to determine the factors responsible for inducing HIV reactivation and inflammation and explore cellular mechanisms that antagonize these factors in order to develop a molecular understanding of the etiology of HAND. Three members of the Nerve Growth Factor IB-like nuclear receptor family, which includes nuclear receptor 77 (Nur77, NR4A1), nuclear receptor related 1 (Nurr1, NR4A2), and neuron-derived receptor 1 (Nor1, NR4A3), are strong candidates for factors that mediate HIV silencing in microglial cells. These receptors play complementary roles in neurons and microglia to limit inflammatory responses. In neurons, these receptors act as positive transcriptional regulators that control expression of dopamine transporter and tyrosine hydroxylase for differentiation of dopamine neuron, as well as other key genes involved in neuronal survival and brain development [38–41]. By contrast, these nuclear receptors can also act as negative transcriptional regulators in microglia cells and suppress expression of inflammatory cytokines such as TNF-α and IL-1β [42]. Because of these combined mechanisms, Nerve Growth Factor IB-like nuclear receptors play a critical role in protection of the brain during neurodegenerative diseases such as Parkinson's disease and Alzheimer's disease [43–48].

Here we report that Nurr1 plays a pivotal role in silencing active HIV in microglial cells by recruiting the CoREST/HDAC1/G9a/EZH2 transcription repressor complex to HIV promoter. Transcriptomic data from cells expressing various levels of Nurr1 additionally demonstrated that it promotes microglial homoeostasis and suppression of inflammation in the brain.

## Results

### Nurr1 agonists strongly induce HIV silencing in microglial cells

To study the role of nuclear receptors in the control of HIV expression in a human cell background, we used our immortalized human microglial (hμglia) cells [18] infected with a recombinant HIV-1 reporter that carried an EGFP marker for "real-time" monitoring of HIV latency and reactivation (**Fig 1A**) [18–21,49]. A hallmark of the HIV transcriptional control in microglia is that the cells quickly revert to latency once the activation stimulus is removed (**Fig 1B and 1C**).

To objectively identify mechanisms regulating HIV expression in microglial cells, we initially performed a global shRNA screen for HIV silencing cellular factors using a comprehensive lentiviral library [50,51]. The latently infected CHME/HIV rat microglial cell line was superinfected with lentiviral vectors carrying a synthetic shRNA library from Cellecta Inc. (Mountain View, CA) containing a total of 82,500 shRNAs targeting 15,439 mRNA sequences [50,51]. Cells carrying reactivated proviruses were then purified by sorting and the shRNA sequences were identified by next-generation sequencing and classified by Ingenuity Pathway Analysis (QIAGEN).

This powerful technology, which we have also applied to the identification of latency factors in T-cells and TB-infected myeloid cells [50,51], has revealed a central role for nuclear receptors in maintaining proviral latency in microglial cells. Analysis of the top 25% "hits" led to our unexpected discovery that the orphan nuclear receptors from the Nerve Growth Factor

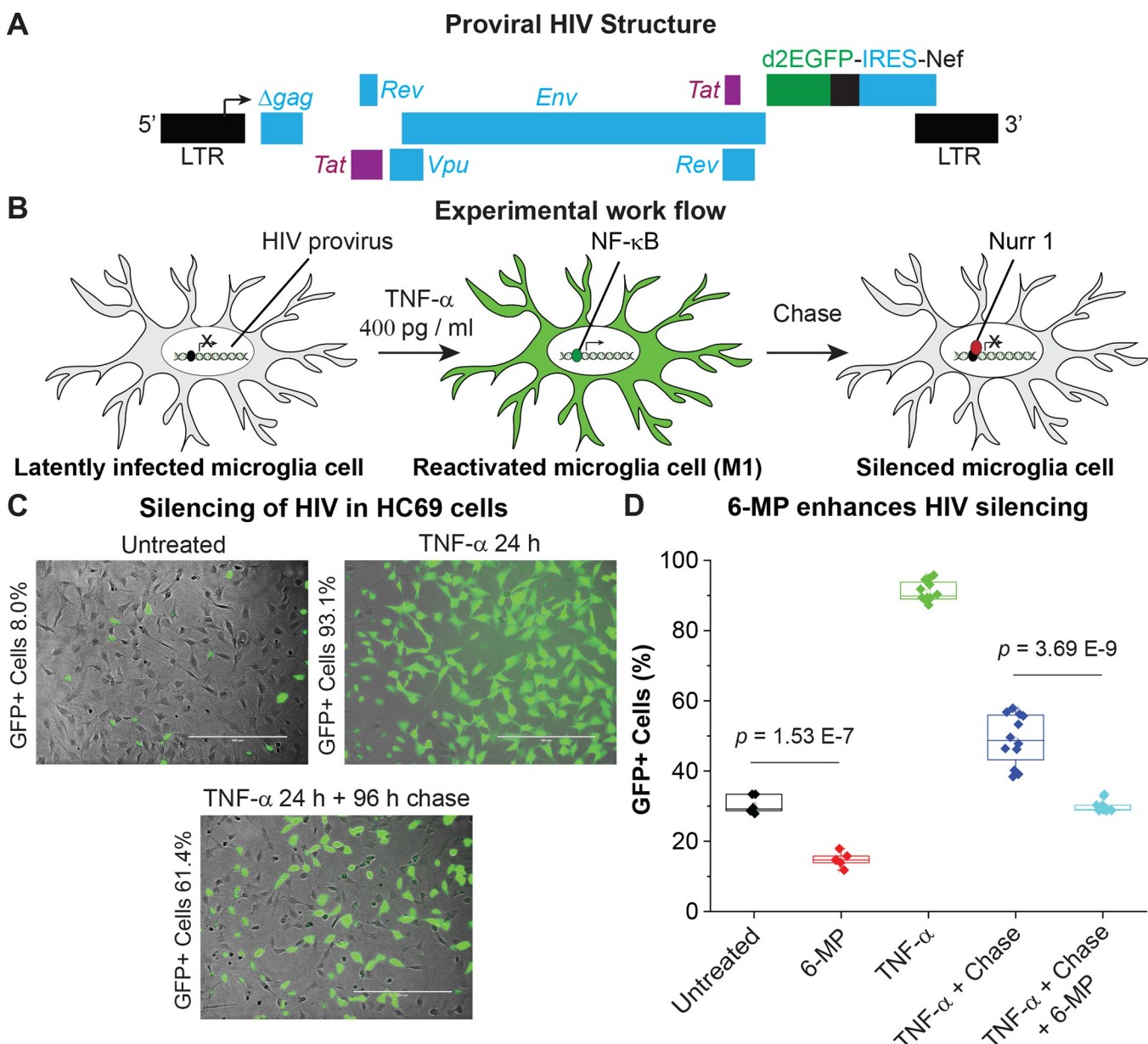

**Fig 1. Accelerated silencing of HIV in microglial cells by the Nurr1/Nor1 agonist 6-MP. A**, genome organization of a d2EGFP reporter HIV-1 that was cloned in the lentiviral vector pHR'. A fragment of HIV-1$_{pNL4-3}$, containing *Tat*, *Rev*, *Env*, *Vpu* and *Nef* with the green fluorescence reporter gene d2EGFP, was cloned into the lentiviral vector pHR'. The resulted plasmid was used to produce the VSV-G HIV particles as described previously [122]. Immortalized human microglial cells (hμglia) were infected with the lenti-HIV viral particles, generating multiple clones with an integrated pro-virus genome. HC69 was a representative of these clones. **B,** Schematic diagram of experimental scheme to study the role of nuclear receptors in microglial reactivation and reversion to latency. **C,** Representative phase contrast, GFP, and overlapped images of HC69 cells that were cultured in the absence (untreated, left panel) and in the presence of TNF-α (400 pg/ml) for 24 hr (TNF-α 24 h, middle panel) respectively, or used in a chase experiment by continuously culturing HC69 cells in the absence of TNF-α for 96 hr after stimulating the cells with TNF-α (400 pg/ml) for 24 hr and washing with PBS (TNF-α 24 h+96 h, right panel). The average percentages of GFP+ cells indicated for each panel were measured by flow cytometry from triplicate wells. **D,** *Impact of Nurr1 agonist* 6-MP on HIV silencing. Quantitative flow cytometry results from 9 independent experiments performed over a period of 2 years are shown. The *p*-values of pair-sample, Student's *t*-tests comparing untreated cells and cells treated with 1 μM 6-MP are indicated.

IB-like family (NR4A1 (Nur77), NR4A2 (Nurr1) and NR4A3 (Nor1)) ranked in the top 25%. Consistent with these results, agonists of the NR4A nuclear receptor family (Nur77 (NR4A1), Nurr1 (NR4A2), and Nor1 (NR4A3)) have been shown to reduce microglial activation and

neuron degeneration in animal models [45,52–55]. The glucocorticoid receptor (GR, NR3C), which we have shown previously silences HIV ranked in the top 15% of the hits [20].

In the absence of an exogenous stimulus most HC69 cells were GFP-negative (GFP⁻), indicating latent infections in a high proportion of the cells (**Fig 1C**). Exposure of HC69 cells to TNF-α (400 pg/ml) for 24 hours (hr), induced GFP expression (GFP⁺) in over 90% of the cells, demonstrating that majority of the integrated HIV provirus was in a latent state under normal culture conditions. Notably, the numbers of GFP+ cells decreased from 93.1% to 61.4% at the end of the chase experiment, suggesting the existence of an intrinsic cellular mechanism that silences the activated HIV. This substantial decrease of GFP+ expression was unlikely to be due to GR-mediated HIV silencing [20], because the cells were cultured in the absence of GR ligand glucocorticoid or dexamethasone.

To study the role of the orphan nuclear receptors in viral silencing we conducted chase experiments by culturing the activated HC69 cells for 96 hr in fresh medium following TNF-α stimulation for 24 hr and washing with PBS in presence or absence of the Nurr1 agonist *6-mercaptopurine* (6-MP) [56,57]. As shown in **Fig 1D**, which is a compilation of data from 9 different experiments performed over a two-year period, 6-MP significantly reduced the frequency GFP+ cells in the absence of an exogenous stimulus. After activation of the cells with TNF-α, more than 91% of the cells in the population became GFP+. Following the chase, the number of GFP+ cells was reduced to 48.9% in the absence of 6-MP, but only 29.8% remained GFP+ in the presence of 1 μM 6-MP.

As shown in **Fig 2A**, the frequency of GFP⁺ cells decreased in a 6-MP dose-dependent manner. Data from Western blot analysis showed that HC69 cells constitutively expressed Nurr1, as well as a low level of Nor1 (**Fig 2B**), but Nur77 expression in these cells was below the detection limit. Treatment with 6-MP slightly increased expression of Nurr1 and strongly induced expression of Nur77 at 5 μM, as reported previously [56,57]. Consistent with the GFP reporter results, expression of HIV, as measured by the levels of Nef protein, was also strongly inhibited by 6-MP in a dose-dependent manner. Notably, as a control for the role of Nurr1 in cellular gene expression, 6-MP also substantially reduced expression of MMP2, which is a well-known repression target of Nurr1 and a neurotoxin involved in the development of HAND [58,59].

To rule out any possible cell line specific clonal effects of Nurr1 on HIV silencing, we infected the originally immortalized human microglial cells C20 [18] with the same EGFP reporter HIV shown in **Fig 1A** and selected five clones that displayed high levels of GFP expression due to spontaneous reactivation (**Fig 2C**). We then cultured the five different clones as well as a mixed population of HIV infected C20 cells in the presence of DMSO (placebo) or 6-MP for two days, and measured Nef expression in the treated cells by Western blot analysis. As shown in **Fig 2C**, the mixed cell population and each of the clones (with the exception of clone#25), exhibited 50 to 60% reductions in Nef expression after 6-MP treatment. Because of this reproducible response, the representative clone, HC69 [18–21], was used for most of the remaining experiments described in this study.

Members of the nuclear receptors (NRs) families including Thyroid Hormone Receptor-like family members PPARα, PPARβ, PPARγ, and RARβ ranked in the top 5%, and the Retinoid X Receptor-like family members RXRα and RXRβ ranked in the top 15%, Since Retinoid X Receptor-like family members also play a critical role in silencing inflammation in the brain [60,61], we also screened various agonists of the nuclear receptors for their effect on HIV expression in HC69 microglia (**S1C Fig**). We induced maximum HIV expression in HC69 cells with a high dose (400 pg/ml) of TNF-α for 24 hr, followed by a chase during which the induced cells were cultured in the absence or presence of various combinations of nuclear receptor agonists. Consistent with our shRNA data, the RXRα/β/γ agonist bexarotene (BEXA) [62–65] silenced HIV expression on its own, although it was less potent than 6-MP.

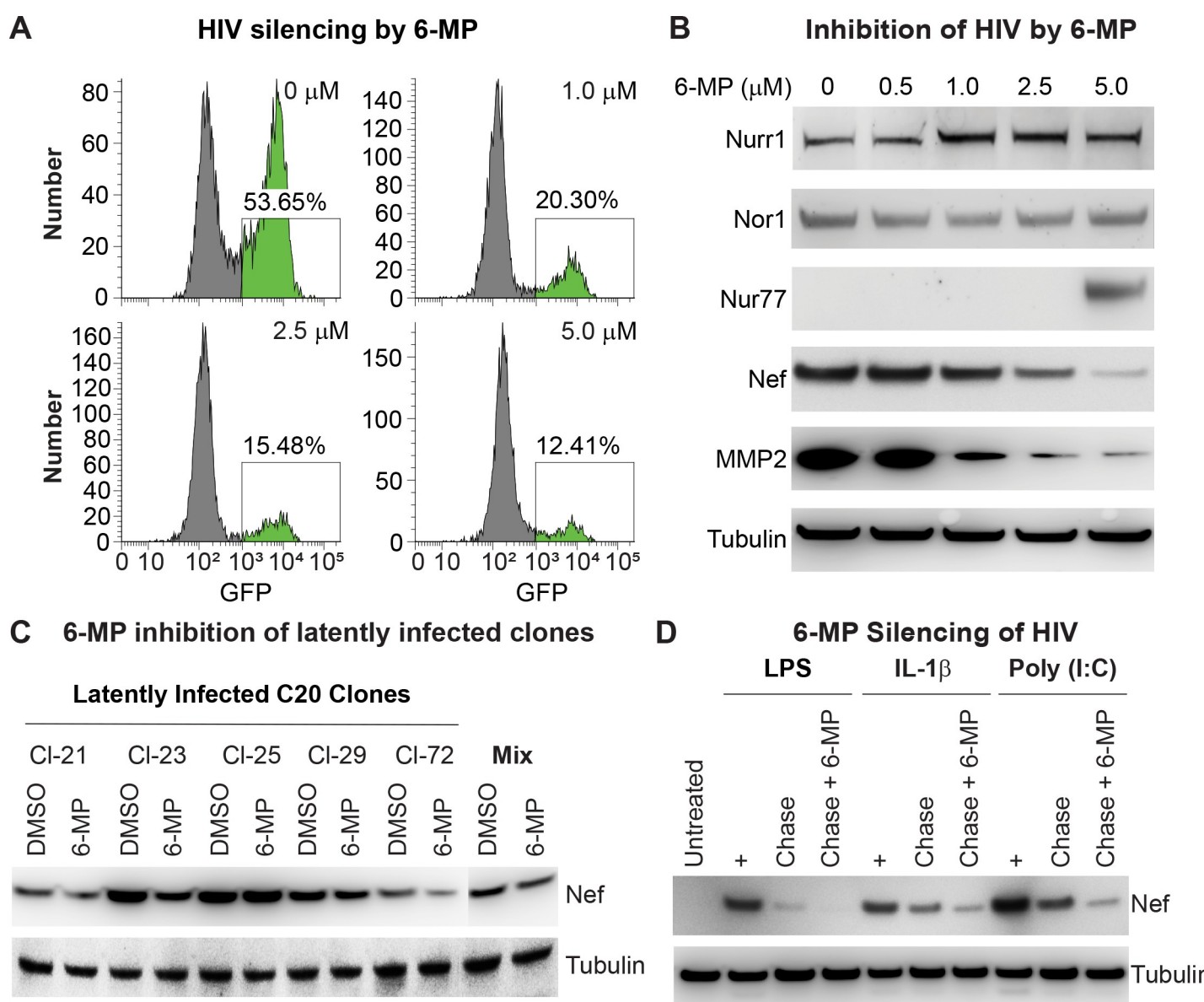

**Fig 2. Inhibition of HIV expression by Nurr1 agonist 6-MP. A**, Representative flow cytometry histograms showing the i*mpact of Nurr1 agonist* 6-MP on HIV expression in HC69 cells in a dose-dependent manner. **B**, Western blot detection of Nur77, Nurr1, Nor1, HIV-1 Nef protein, and Nurr1 target gene MMP2 in HC69 cells. Proteins were detected using a rabbit polyclonal anti-Nurr1 antibody (Sata Cruz Biotechnology, Cat# sc-991), a mouse monoclonal anti-Nor1 antibody (Perseus Proteomics, Cat# PP-H7833-00), a rabbit polyclonal anti-Nur77 antibody (Cell Signaling, Cat# 3960S), a mouse monoclonal anti-HIV Nef antibody (Abcam, Cat#ab42355), and a rabbit monoclonal anti-MMP2 antibody (Cell Signaling, Cat#40994). The level of β-tubulin was used as a loading control. 6-MP at 5 μM strongly induced Nur77 expression, which is consistent with a previous report [123] but did not effect Nurr1 or Nor1 expression. **C**, Western blot detection of Nef expression from individual clones derived from HIV-infected immortalized human microglial cell (C20) [18] using the pHR' HIV reporter. The right panel shows the parental mixed population of HIV infected C20 cells. Cells were cultured in the presence of DMSO (placebo) or 6-MP (1 μM) for 48 hr. β-Tubulin was used as a loading control. **D**, Western blot measurement of Nef protein from HC69 cells that were untreated or induced with LPS (1 μg/ml), IL-1β (50 pg/ml), and poly (I:C) (PIC, 500 ng/ml) for 24 hr. The cells were chased in the absence or presence of 1 μM 6-MP for 72 hr after stimulation. β-Tubulin was used as loading control.

Interestingly, combinations of 6-MP with DEXA and BEXA displayed additive HIV silencing effects, suggesting that they have complementary and distinct mechanisms of action. Since the cytotoxicity of 6-MP and the nuclear receptor agonists was minimal at the used concentrations (**S1A Fig**), their effects on HIV silencing cannot be ascribed to low cell viability following treatment.

To test if 6-MP promotes silencing of active HIV independently of the initial induction signal, we stimulated HC69 cells with lipopolysaccharide (LPS), IL-1β, or poly (I:C) (PIC) for 24 hr and then cultured the cells in the absence or presence of 6-MP for 48 hr. Data from Western blot analysis indicated that 6-MP reduced the levels of HIV Nef induced by each of these activators by at least 70% (**Fig 2D**).

## Nurr1 overexpression enhances HIV silencing

To further examine how the nuclear receptors contribute to HIV silencing, we constructed lentiviral vectors expressing either N-terminal 3X-FLAG-tagged Nur77, Nurr1, or Nor1 under the control of a CMV promoter. Infection of HC69 cells with the different lentiviruses generated cell lines that stably expressed either FLAG-tagged Nur77, Nurr1, Nor1, or the empty vector (**Fig 3**). Overexpression of Nurr1 was rigorously confirmed by mapping of normalized RNA-Seq reads to the Nurr1 gene locus (**Fig 3A**). The overexpression of each of the NR4A1 (Nur77), NR4A2 (Nurr1) and NR4A3 (Nor1) nuclear receptors was further verified by western blots (**Fig 3B**).

To examine how overexpression of each of these nuclear receptors modulates HIV proviral silencing after reactivation, chase experiments were performed following TNF-α stimulation (**Fig 3C**). In this protocol, we stimulated all four cell lines with a high dose (400 pg/ml) of TNF-α for 24 hr to induce HIV transcription through activation of NF-κB [19], followed by a 48 hr chase during which TNF-α was removed by washing the cells with PBS followed by the addition of media lacking TNF-α (**Fig 3C**). As shown by the western blot in **Fig 3D**, TNF-α strongly induced the expression of HIV Nef protein, which we used as a marker of HIV reactivation, in all cell lines at 24 hr. Notably, Nef expression decreased in all four cell lines during the 48 hrs after TNF-α withdrawal. However, the reduction in Nef expression was much more pronounced in HC69 cells that expressed 3X-FLAG-Nurr1 than any of the other receptors, suggesting that overexpression of Nurr1 selectively enhances silencing of active HIV in HC69 cells.

We additionally confirmed the western blot data using RNA-Seq (**Fig 3E**), which permitted us to accurately and simultaneously measure the fluctuations in both HIV and Nurr1 mRNA expression. In the Nurr1 overexpressing cells, in unstimulated conditions, the basal level of HIV proviral expression was 1.8-fold higher than cells expressing empty vector. Following stimulation with either a low dose (20 pg/ml), or high dose (400 pg/ml), of TNF-α, both vector-infected and Nurr1 overexpressing cells showed an increase in proviral expression. While the level of HIV expression was similar between control cells (vector-infected) and Nurr1-overexpressing cells after high dose TNF-α stimulation, Nurr1 overexpressing cells had much lower proviral expression level after low dose TNF-α stimulation (**Fig 3E**). The level of HIV mRNA after withdrawal of high dose TNF-α was three times lower in Nurr1 overexpressing cells than in vector-infected cells (**Fig 3E**), strongly suggesting that overexpression of Nurr1 enhanced silencing of active HIV in HC69 cells.

As a control for clonal variation, we also generated cell lines from clone#29 stably infected with either the empty 3X-FLAG-lentivirus (vector) or the lentivirus expressing 3X-FLAG-Nurr1 (**S2A Fig**). Chase experiments performed using these cells confirmed that Nef expression is reduced by more than 50% in cells overexpressing Nurr1 compared to control cells (vector) at the end of the chase experiment (**S2A Fig**).

## Nurr1 knockdown blocks HIV silencing

As a complementary approach, we also performed shRNA-mediated knock down (KD) of endogenous Nurr1 in HC69 cells. Cell lines that stably expressed Nurr1-specific, or control

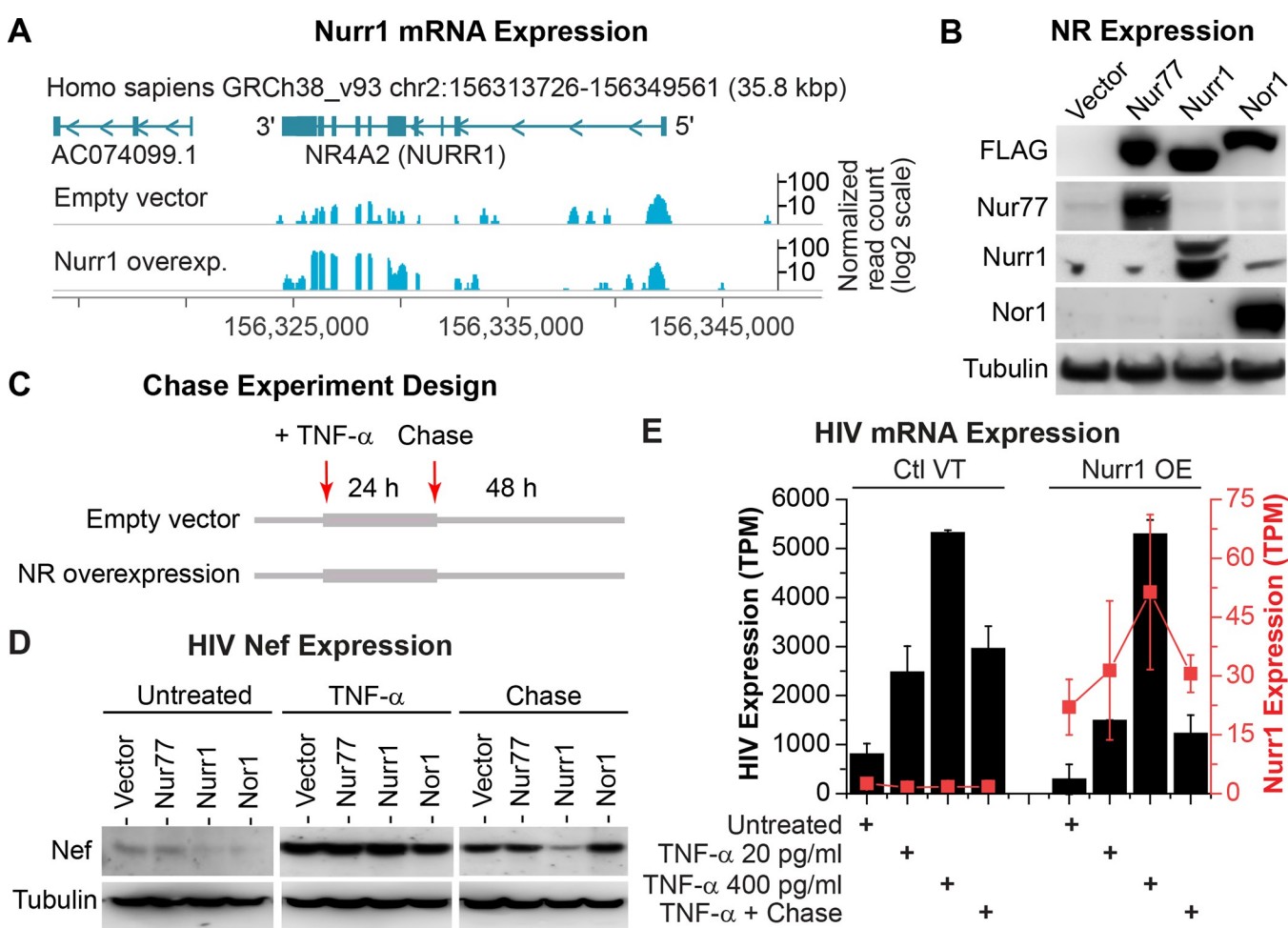

**Fig 3. Overexpression of Nurr1 in HC69 cells enhances HIV silencing. A,** RNA-Seq confirmation of overexpression (OE) of Nurr1 in HC69 cells. Sequence read histograms for the Nurr1 locus is shown for control (vector) and Nurr1 overexpression. Annotated genes for the shown locus are indicated on the top, and the position of the locus on chromosome 2 is shown both at the top and the bottom. A read scale for each row is shown on the right, with the values for the overexpression studies drawn on a log2 scale. **B,** Verification of Nur77, Nurr1, and Nor1 overexpression by Western blot analysis in HC69 cell lines stably expressing 3X-FLAG-tagged Nur77, Nurr1, and Nor1 respectively by using a mouse monoclonal anti-FLAG M2 antibody (Sigma, Cat# F1804) and the nuclear receptor specific antibodies described in **Fig 2B**. HC69 cells stably carrying the 3X-FLAG-empty vector were used as a reference for comparison. The level of β-tubulin was used as a loading control. Notably, the levels of endogenous Nur77 and Nor1 in HC69 cells were very limited. In contrast, Nurr1 was constitutively expressed in HC69 cells. **C,** Schematic depicting the TNF-α stimulation and chase studies. The four cell lines described in **B** were either untreated or treated with high dose (400 pg/ml) TNF-α for 24 hr. To examine HIV silencing, one set of TNF-α induced cells were used in a chase experiment by continuous culture of the cells in the absence of TNF-α for an additional 48 hr. The time points at which TNF-α is added or removed are shown by arrows on the top. **D,** Expression of HIV Nef protein in the different cell lines before and after TNF-α stimulation and at the end of the chase experiment was measured by Western blot analysis. The level of β-tubulin was used as a loading control. **E,** Expression level of HIV mRNA (black bar graph) and Nurr1 (red rectangles and lines) in transcripts per million cellular transcripts are shown for each of the treatment steps shown in panel **C** in both vector-infected cells (on the left) and Nurr1 overexpressing cells (on the right half of the graph). For the 24 hr TNF-α stimulation step, both a low dose (20 pg/ml) and a high dose (400 pg/ml) are used. The values shown are the average of three replicate RNA-Seq samples with two standard deviations as error bars. The expression values for HIV and Nurr1 are shown on Y axes to the left and right, respectively.

(scrambled), shRNA were verified for effective Nurr1 KD by RNA-Seq analyses (**Fig 4A**). Following the protocol described in **Fig 4B**, the control and Nurr1 KD cells were activated with a high dose (400 pg/ml) of TNF-α for 24 hr, followed by a 72 hr chase. Western blot analyses also confirmed the Nurr1 knock down efficiency (**Fig 4C**). The blots also showed that HIV Nef protein, was strongly induced at 24 hr post TNF-α stimulation in both the control and the Nurr1 KD cells. As expected, after the chase, Nef levels decreased significantly in the control

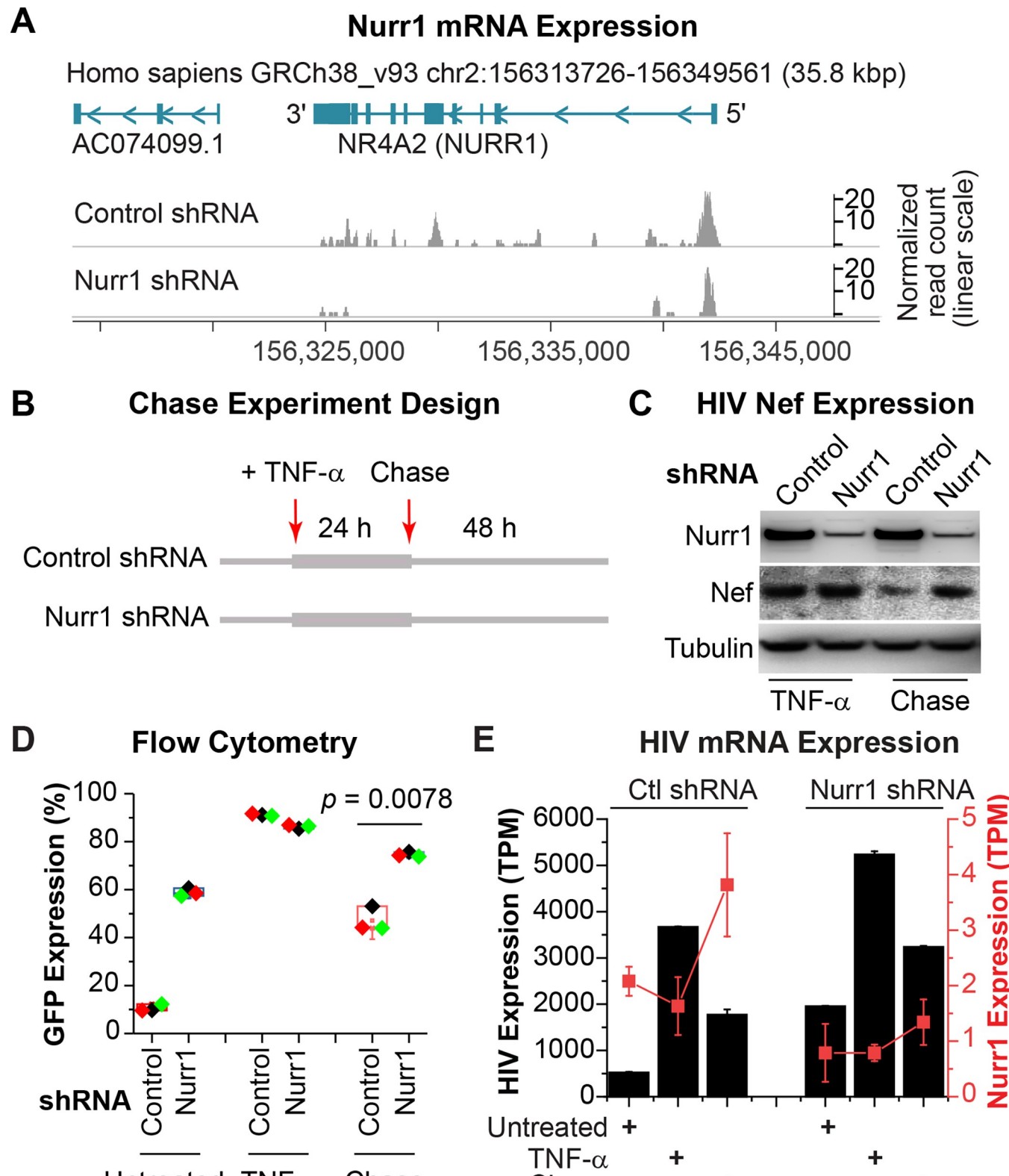

**Fig 4. Nurr1 knock down (KD) in HC69 cells enhances HIV expression and block proviral silencing during the chase step. A,** RNA-Seq confirmation of Nurr1 KD in HC69 cells. Read histograms for the Nurr1 locus is shown for non-targeting shRNA-infected cells, and cells infected with Nurr1 specific shRNA lentiviral constructs. The Nurr1 shRNAs resulted in a 2.6-fold reduction in Nurr1 mRNA level in the Nurr1 KD cells. Annotated genes for the shown locus are

indicated on the top, and the position of the locus on chromosome 2 is shown both at the top and the bottom. A read scale for each row is shown on the right, with the values for the knock down studies drawn on a linear scale. **B,** Schematic depicting the TNF-α stimulation and chase studies. The two shRNA lentiviral transduced cell lines described in **A** were either untreated or treated with high dose (400 pg/ml) TNF-α for 24 hr. One set of TNF-α induced cells were used in a chase experiment in the absence of TNF-α for an additional 48 hr. The time points at which TNF-α is added or removed are shown by arrows on the top. **C,** Western blot studies measuring the expression of endogenous Nurr1, Nef, and β-tubulin in cells infected with either a non-targeting control shRNA or Nurr1-specific shRNA lentiviral vectors. The expression patterns from the TNF-α (400 pg/ml) stimulation and the chase step are shown. **D,** KD of endogenous Nurr1 strongly inhibits HIV silencing. The percentages of GFP$^+$ cells in the two cell lines, before treatment, at 24 hr post-TNF-α (400 pg/ml) stimulation, and at 72 hr after TNF-α withdrawal (chase) were analyzed by flow cytometry and calculated from three independent experiments. The difference in GFP expression between the two cell lines at 72 hr chase was statistically significant, with a $p$ = 0.0078. **E,** Expression level of Nurr1 (red rectangles and lines) and the HIV provirus (black bar graph) in transcripts per million cellular transcripts are shown for each of the treatment steps in both non-targeting shRNA infected cells (on the left) and Nurr1-specific shRNA-infected cells (on the right half of the graph). The values shown reflect the average of three replicate RNA-Seq samples from two distinct shRNA constructs per control and Nurr1 knock down groups, with two standard deviations as error bars. The expression values for HIV and Nurr1 are shown on Y axes to the left and right, respectively.

cells due to expression of the endogenous Nurr1 but remained high in the Nurr1 KD cells (**Fig 4C**).

Similar results were obtained using flow cytometry (**Fig 4D**). Compared to cells expressing control shRNA with 10.5% GFP+ cells, the Nurr1 KD cells displayed 58.8% GFP$^+$ cells even before TNF-α stimulation, which most likely resulted from failure of silencing spontaneously reactivated HIV in these cells due to Nurr1 depletion (**Fig 4D**). As expected, after exposure to a high dose of TNF-α for 24 hr, both the control and Nurr1 KD cell lines expressed equally high levels of GFP expression, displaying 86.3% and 91.2% GFP+ cells respectively. However, 72 hrs after TNF-α withdrawal, GFP expression decreased significantly in cells expressing the control shRNA (47.2% GFP+) but remained high (74.6% GFP+) in the Nurr1 KD cells (**Fig 4D**). Finally, the overall mRNA level of the HIV measured by RNA-Seq was 1.7-fold higher in Nurr1 KD at the end of the chase experiment (**Fig 4E**).

Thus, both the overexpression and the reciprocal KD experiments confirmed an essential role of Nurr1 in the silencing of HIV in microglial cells.

## Nurr1 binding to HIV 5'LTR is essential for HIV silencing

Like most transcription factors, Nurr1 binds to specific DNA motif for transcriptional regulation of its target genes [66]. The orphan and ligand-mediated nuclear receptors form dimers on their target DNAs via highly cooperative assembly of their DNA-binding domains. We identified a putative Nurr1 binding site overlapping the COUP/AP-1 sites in the U3 region of the HIV-1. The site contained an octanucleotide with a canonical nuclear receptor binding motif (NBREP: AAAGGTCA), and across the dyad axis, the IR5 motif which permits binding of either the Nurr1 homodimer or binding of a heterodimer with other nuclear receptors (**Fig 5A**) [66].

Data from ChIP-seq experiments provided direct evidence for Nurr1 recruitment to the proviral LTR in untreated HC69 cells (**S2B Fig**). As expected, Chip assays also showed that there were increased levels of Nurr1 at the HIV 5'LTR in GFP- cells compared to GFP+ cells at 48 hr after TNF-α withdrawal in a chase experiment (**S2B Fig**).

To confirm that Nurr1 binding to HIV is due to direct DNA recognition we generated a DNA binding defective mutant by converting C280 and E281 in the Nurr1 DNA binding domain to alanine (A) (**Fig 5B**) [42]. As shown in **Fig 5C**, ChIP assays showed that the level of Nurr1 at the LTR was increased approximately 2-fold in cells that over expressed the wild type Nurr1 compared to cells expressing either the empty vector or the mutant Nurr1.

The behavior of HC69 cells stably expressing the empty vector, 3X-FLAG-Nurr1 WT, and 3X-FLAG-Nurr1 mutant was then compared in chase experiments following the protocol described in **Fig 3C**. As shown in **Fig 5D and 5E**, the reduction in GFP and Nef expression after the chase was much more pronounced in cells overexpressing the wild type Nurr1,

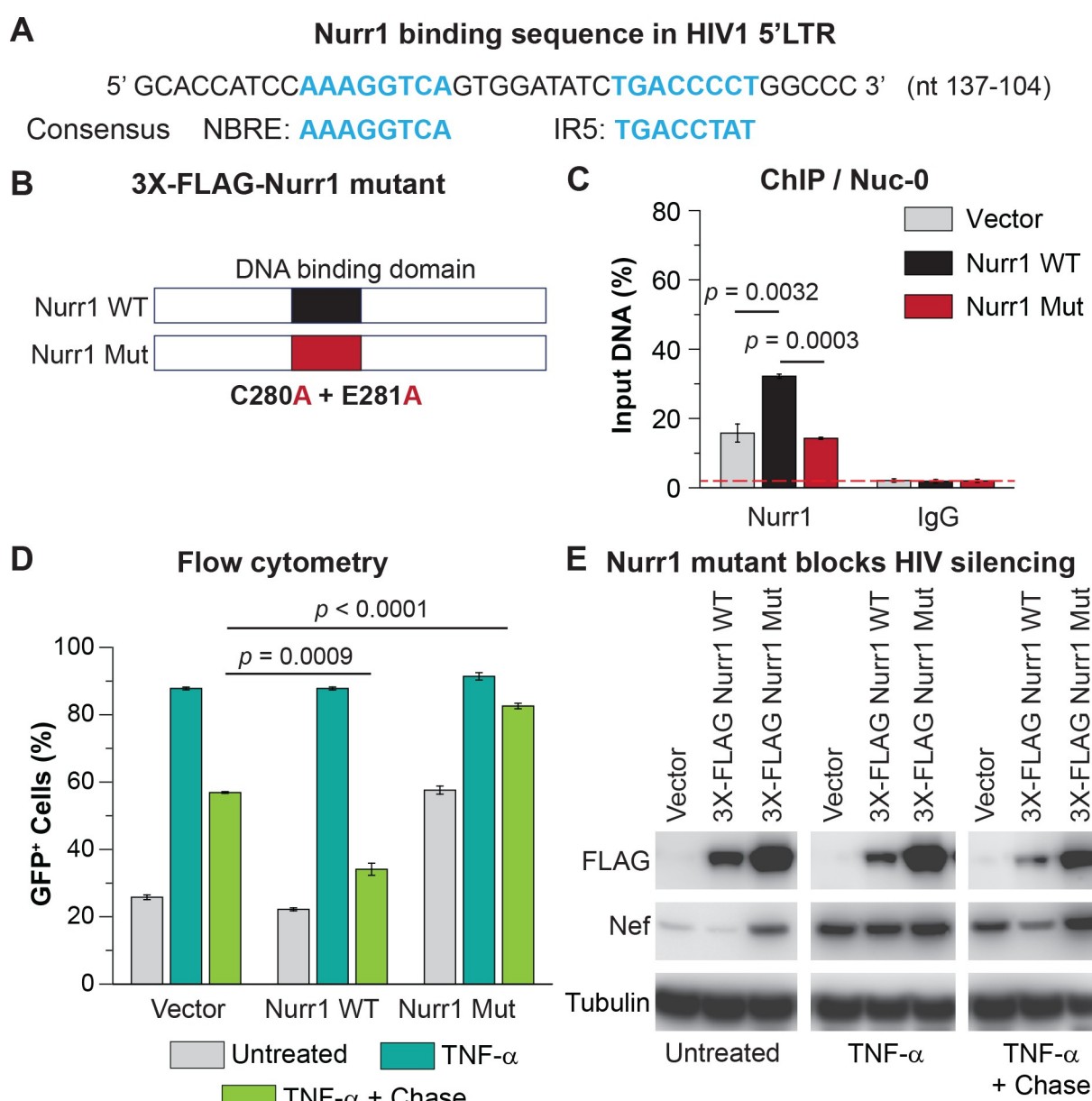

**Fig 5. Nurr1 binds to conserved sites in the HIV LTR. A**, Location of the putative Nurr1 binding site in the HIV LTR consists of a 8-nt NGFI-B-responsive element (NBRE) and an inverted 8-nt repeat (IR5), to which Nurr1 monomers, homodimers, and heterodimers bind [66]. **B**, Lentiviral construct expressing the DNA binding deficient 3X-FLAG-Nurr1 mutant (Nurr1 Mut) was generated from the original lentiviral construct expressing 3X-FLAG-Nurr1 wild type (Nurr1 WT) by converting both residues C280 and E281 to alanine (A) as reported previously [42]. **C**, ChIP assay measurement of the levels of Nurr1 protein at Nuc-0 in the HIV-1 LTR in HC69 cells expressing empty vector (grey bars), 3X-FLAG-Nurr1 WT (black bars), or 3X-FLAG-Nurr1 Mut (red bars). The mean levels of Nurr1 in 5'LTR, presented as the percentage (%) of ChIP product over input DNA of each sample, was calculated from triplicates qPCR of a single ChIP experiment using primers for the Nuc-0 region. **D**, Impact of overexpression of Nurr1 carrying DNA-binding defective mutations on GFP expression. HC69 cells expressing empty vector, 3X-FLAG-Nurr1 WT, or 3X-FLAG-Nurr1 Mut were induced with a high dose (400 pg/ml) TNF-α for 24 hr and then chased in the absence of TNF-α for 48 hr and the fraction of GFP+ cells was measured by flow cytometry. The *p*-values of pair-sample, Student's *t*-tests for differences in the numbers of GFP+ cells between the control cells (Vector) and cells expressing wild-type Nurr1 (Nurr1 WT) or between the control cells (Vector) and cells expressing mutant Nurr1 (Nurr1 Mut) at the end of chase were calculated from three technical repeats. **E**, Impact of overexpression of Nurr1 carrying DNA-binding defective mutations on HIV Nef expression. Western blots were performed on HC69 cells expressing empty vector, 3X-FLAG-Nurr1 WT, or 3X-FLAG-Nurr1 Mut using antibodies directed against FLAG, Nef or β-Tubulin, which served as a loading control.

compared to the control vector. In contrast, in cells overexpressing the mutant Nurr1, expression of both GFP and Nef remained high after TNF-α withdrawal, demonstrating that Nurr1 DNA binding is essential for HIV silencing.

It is important to note that the overall cellular levels of Nurr1 protein do not correlate with the levels of Nurr1 at the HIV LTR. In fact, cellular Nurr1 protein levels increase slightly after TNF-α stimulation and were higher in the GFP+ cells than in the GFP- cells at the end of chase (S2C Fig). Instead, the decrease in Nurr1 at the LTR appears to be caused by gene-specific degradation due to sumoylation of Nurr1 upon TNF-α stimulation (S2D Fig), consistent with previous reports [42].

## Nurr1 promotes the recruitment of the CoREST/HDAC1/G9a/EZH2 repressor complex to the HIV promoter

Nurr1 binding to HIV provirus may inhibit HIV transcription either through direct repression or by engaging a trans-repression mechanism involving the recruitment of specific transcription repressors. A Nurr1-mediated trans-repression mechanism involving the recruitment of the CoREST transcription repressor complex regulates inflammatory cytokine synthesis, such as TNF-α and IL-1β in microglial cells, following LPS stimulation [42,67]. CoREST can interact in a dynamic manner with multiple epigenetic silencing machinery components including histone deacetylases 1/2 (HDAC1/2), euchromatic histone lysine N-methyltransferase 2 (G9a; EHMT2), lysine (K)-specific demethylase 1A (KDM1A), and enhancer of zeste 2 polycomb repressive complex 2 subunit (EZH2) [68,69] (Fig 6A).

To evaluate the role of CoREST in HIV silencing in microglial cells, we first conducted co-immunoprecipitation (Co-IP) assays to confirm the association of Nurr1 with components of the CoREST repressor complex in HC69 cells (Fig 6B). HC69-3X-FLAG-vector and HC69-3X-FLAG-Nurr1 cells were treated with and without a high dose (400 pg/ml) of TNF-α for either 4 hr or 24 hr. After 24 hr of TNF-α treatment, the cells were chased in the absence of TNF-α for a further 24 hr. Total protein lysates from the differently treated cells were immunoprecipitated using a mouse monoclonal anti-FLAG antibody conjugated to magnetic beads. The anti-FLAG beads pulled down not only FLAG-tagged Nurr1 but also CoREST, HDAC1, G9a, and EZH2 from the HC69-3X-FLAG-Nurr1 cell lysates, demonstrating that in the microglial cells Nurr1 bound directly to the CoREST repressor complex. Notably, the amount of CoREST associated with Nurr1 increased after the cells were stimulated with TNF-α. In contrast, the amounts of G9a and EZH2 proteins associated with Nurr1 decreased at 4 hr post-TNF-α stimulation but rebounded at 24 hr post-TNF-α stimulation. Together, these results suggested that the Nurr1/CoREST/HDAC1/G9a/EZH2 complex were most likely dissociated from each other during early time points of TNF-α stimulation but were reassembled at later time points.

To provide direct evidence that Nurr1 mediates the recruitment of the CoREST/HDAC1/G9a/EZH2 complex to HIV promoter, we treated HC69 cells expressing control shRNA and Nurr1 shRNA with high dose TNF-α, followed by a 24 hr chase. We then conducted additional ChIP experiments and measured the ChIP products by quantitative PCR (qPCR). As shown in Fig 6C, CoREST was strongly recruited to HIV promoter at 4 hr post TNF-α stimulation in HC69 cells expressing control shRNA, however, its recruitment was substantially inhibited in Nurr1 KD cells. Similarly, G9a and HDAC1 levels in HIV promoter peaked at 24 hr post TNF-α stimulation in HC69 cells expressing control shRNA but their recruitment was also reduced in Nurr1 KD cells (Fig 6D).

A higher resolution analysis of the recruitment of Nurr1 and the CoREST complex to the HIV provirus was obtained by ChIP-Seq experiments (Fig 7). Nurr1, CoREST, EZH2 and G9a

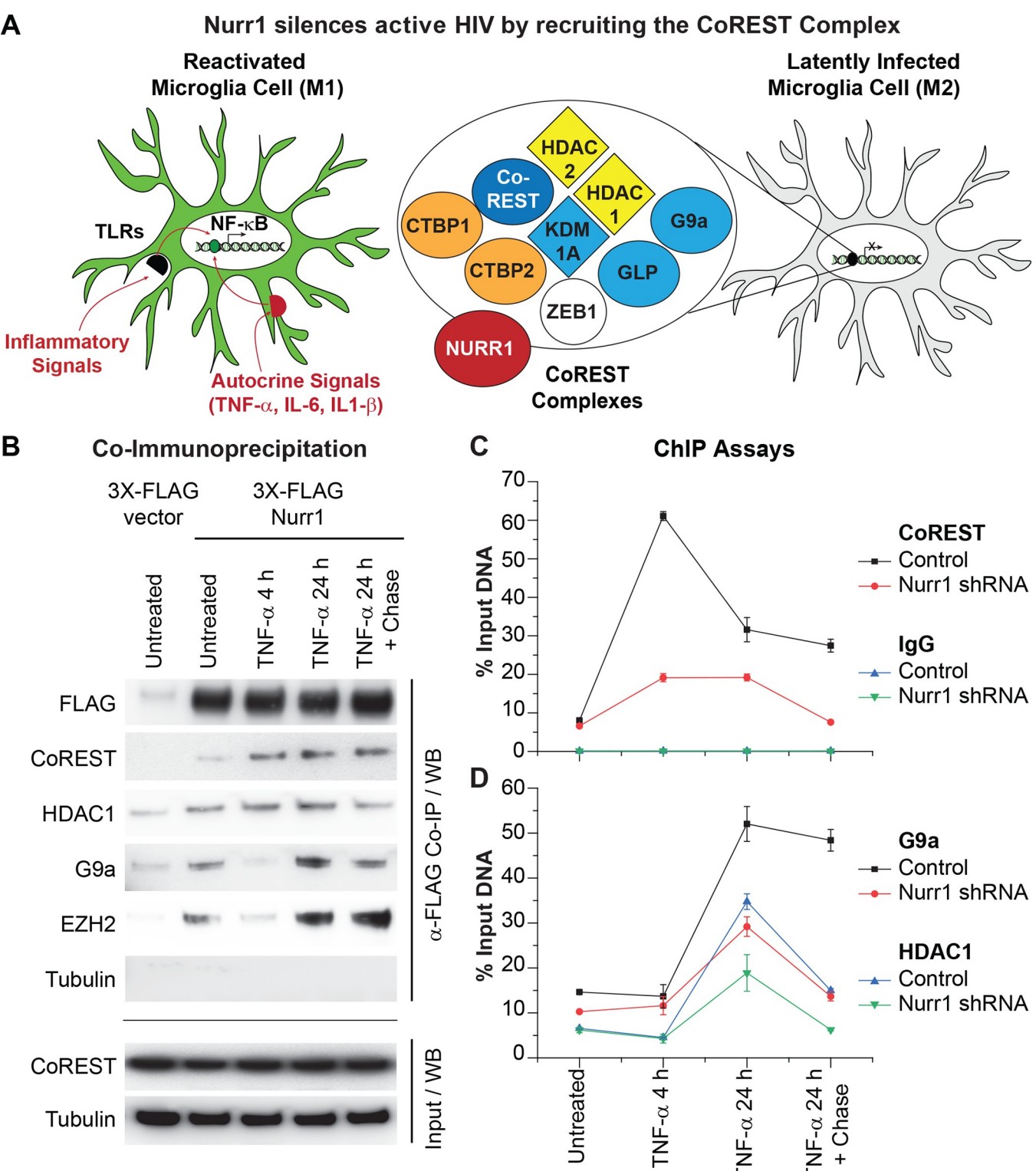

**Fig 6. Nurr1 promotes recruitment of the CoREST repressor complex to HIV promoter. A,** Schematic illustration of Nurr1-mediated epigenetic silencing of active HIV in microglial cells by recruiting the CoREST/HDAC1/G9a/EZH2 repression complex to HIV promoter. **B,** Nurr1 associates with CoREST, HDAC1, G91, and EZH2 to form a transcription repression complex in microglial cells (HC69). HC69-3X-FLAG-vector and HC69-3X-FLAG-Nurr1 cells were cultured in the absence (untreated) or presence of high dose (400 pg/ml) TNF-α for 4 hr and 24 hr respectively. A portion of these cells were also used in a chase experiment by culturing the cells for an additional 24 hr (chase) after stimulation with high dose TNF-α for 24 hr and subsequent washing with PBS (TNF-α 24h+24h). Total protein lysates from the differently treated cells were isolated and used for co-immunoprecipitation (Co-IP) with a mouse anti-FLAG

monoclonal antibody. The original protein lysates (Input) and the Co-IP products were analyzed by Western blot analysis with antibodies to FLAG, CoREST, HDAC1, G9a, EZH2, and β-tubulin respectively. **C**, levels of CoREST (**Top**) and G9a (**Bottom**) at HIV Nuc1 (+30 to +134) in HC69-control shRNA (Control) and HC69-Nurr1 shRNA (Nurr1 KD) cell lines. Cell were activated with TNF-α and chased as described in **B**. The levels of CoREST and G9a in HIV 5'LTR were measured by qPCR and calculated as percentages of the amounts of ChIP products over input DNA from three technical replicates.

were present near the promoter region of the HIV provirus in untreated cells, but there were only low levels of HDAC1. Each factor showed unique kinetics following TNF-α activation and the subsequent chase. For example, following TNF-α activation for 24 hrs there was a precipitous loss of Nurr1 and an increase of HDAC1, consistent with maximal HIV transcription. After the chase, when latency was restored, Nurr1 levels increased and HDAC1 levels decreased. The levels of CoREST at the HIV promoter peaked at 4 hr post-TNF-α stimulation and then declined to basal levels by 24 hr and after the chase. By contrast the levels of G9a and EZH2 at the HIV promoter peaked at 24 hr post-stimulation and remained high after the chase. Thus, there is a dynamic exchange between the epigenetic silencing factors and CoREST.

## Epigenetic silencing of HIV in microglial cells

To confirm the functional role of the histone modifying machinery recruited by Nurr1 and CoREST to the HIV promoter, we conducted additional ChIP assays to monitor the expected histone modifications. As shown in **Fig 8**, higher levels of repressive histone methylation marks H3K27me3 and H3K9me2 but lower levels of acetylated histone mark H3K27Ac and RNA polymerase II (RNAP II) were detected in HC69-3X-FLAG-Nurr1 cells than in HC69-3X-FLAG-vector cells at the end of 48 hr chase after TNF-α stimulation for 24 hr, which is fully consistent with the rapid HIV silencing in the Nurr1 overexpression cells shown earlier (**Fig 3D**).

To further investigate how the CoREST/HDAC1/G9a/EZH2 complex contributes to HIV silencing, we treated HC69 cells with high dose (400 pg/ml) of TNF-α for 24 hr followed by a chase in the absence or presence of epigenetic inhibitors that target the CoREST complex, specifically: HDAC inhibitor suberoylanilide hydroxamic acid (SAHA), G9a inhibitor UNC0638, and EZH2 inhibitor GSK343 (**Fig 9A**). The activity of each of these inhibitors in inducing relevant histone modifications and HIV expression in T cells has been extensively examined [70], and they had minimal toxicity in HC69 cells (**S1A Fig**).

As described previously the levels of GFP+ cells dropped from 88.4% to 67.03% during the chase when cells were cultured in the absence of the inhibitors (**Fig 9A**). However, in the presence of SAHA, UNC0638, or GSK343, the numbers of GFP+ cells remained higher (i.e., 77.8%, 85.5%, and 84.7% respectively), indicating that functional inhibition of these epigenetic silencers prevented active HIV from reverting to latency. All three inhibitors also induced HIV reactivation in latently infected HC69 cells (**S1B Fig**), thus underscoring the importance of these epigenetic mechanisms for HIV silencing in microglial cells.

To confirm the role of these epigenetic silencers, we generated HC69 cell lines stably expressing CoREST-specific shRNA or CRISPR/Cas9/guide RNA (gRNA) for G9a or EZH2. We confirmed successful KD or knock out (KO) of these proteins in these cell lines by Western blot analysis (**Fig 9B** and **9C**). The genetically modified cells were activated with a high dose (400 pg/ml) of TNF-α for 24 hr, followed by culturing the cells in the absence of TNF-α for 48 hr and measurement of GFP expression. CoREST KD substantially increased GFP expression (80.1% GFP+ vs. 25.8% in control cell) even without TNF-α stimulation (**Fig 9D**). Stimulation with high-dose TNF-α for 24 hr resulted in 94.1% and 84.9% GFP+ cells in CoREST KD and control cells respectively. However, after TNF-α withdrawal and subsequent culture for 48 hr,

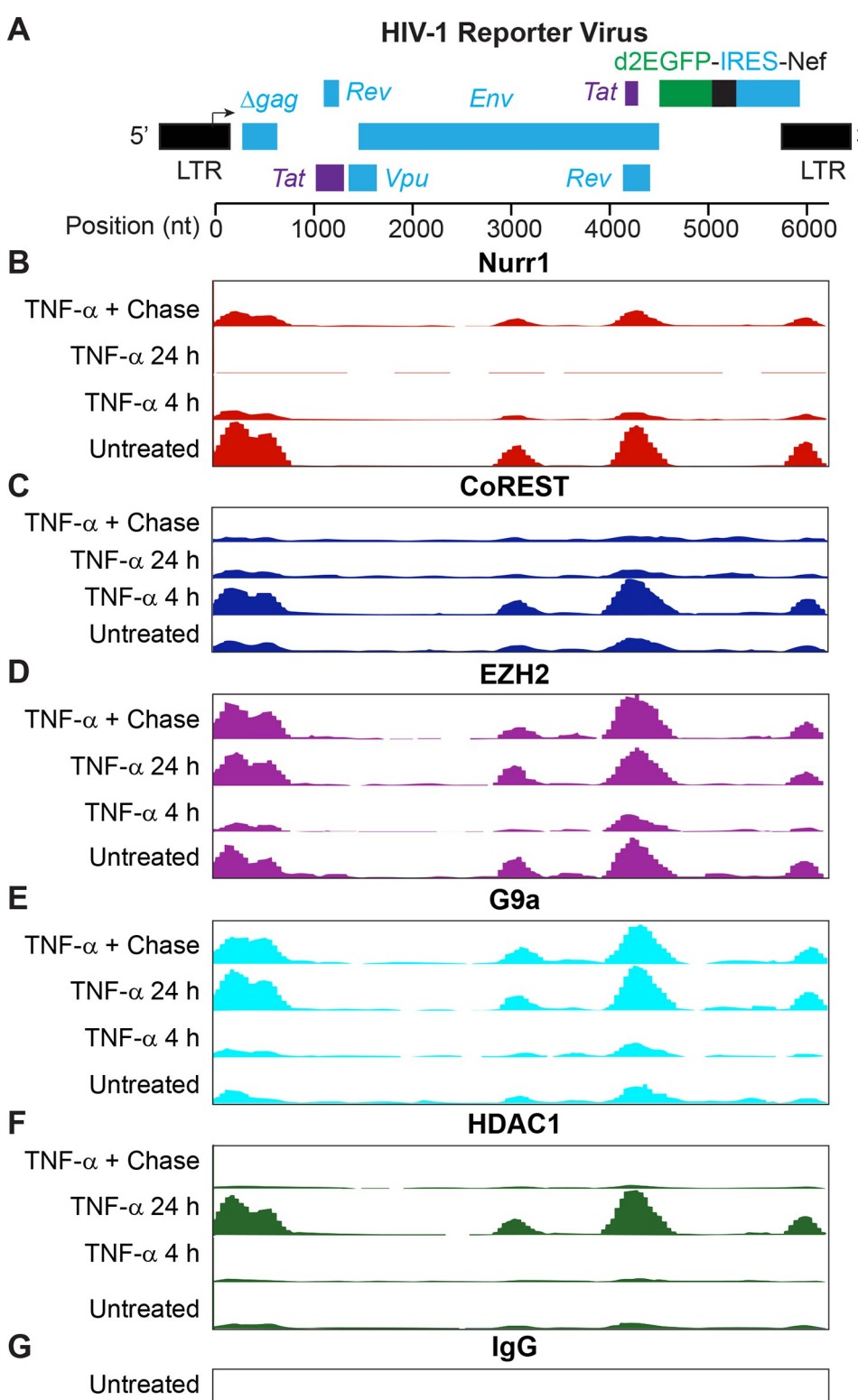

**Fig 7. ChIP-Seq analysis of the recruitment of Nurr1 and histone modifiers to the HIV provirus.** Histograms show numbers of sequence reads on the Y axis along the length of the reporter HIV-1 pro-viral genome on the X axis. A, Aligned reporter genome. B, Nurr1 (Santa Cruz Biotech, Cat #sc-81345). C, CoREST (Cell Signaling, Cat #14567. D, EZH2 (Cell Signaling, Cat #5246S). E, G9a (Cell Signaling, Cat #3306S). F, HDAC1 (Santa Cruz Biotechnology, Cat #sc-7872). G, IgG control. For each antibody, chromatin was prepared from HC69 cells that were untreated, induced

with TNF-α (400 pg/ml) for 4 hr and 24 hr respectively, or used in a chase experiment by continuously culturing HC69 cells in the absence of TNF-α for 24 hr after stimulating the cells with TNF-α (400 pg/ml) for 24 hr. Construction of ChIP-Seq DNA libraries with the ChIP products, enrichment for HIV-1 specific sequences, and data analysis following Ion Torrent sequencing were described in Materials & Methods. The sequence reads on the Y axis were set at the same scales for different time points of ChIP-seq using the same antibody but varied for ChIP-seq with different antibodies, which were from 0 to 2800 reads for ChIP-Seq/Nurr1, 0 to 200 reads for ChIP-Seq/EZH2, 0 to 400 reads for ChIP-Seq/CoREST, 0 to 300 reads for ChIP-seq/G9a, 0 to 1300 reads for ChIP-Seq/HDAC1, and 0 to 200 reads for ChIP-Seq/IgG, respectively.

the numbers of GFP⁺ cells decreased significantly in cells expressing control shRNA (67.7%) but remained high in CoREST KD cells (91.3%), confirming that CoREST was crucial for the silencing of active HIV in microglial cells. Similar results were seen with the G9a and EZH2 KO cell lines (**Fig 9E**).

Therefore, both the ChIP experiments and gene knockout results demonstrate a pivotal role for the CoREST/HDAC1/G9a/EZH2 transcription repressor complex in silencing active HIV in microglial cells. Taken together, these results clearly demonstrated a pivotal role for Nurr1 in mediating recruitment of the CoREST/HDAC1/G9a/EZH2 machinery to the promoter of active HIV for epigenetic silencing.

## Regulation of Nurr1 trans-repression of HIV-1 transcription by phosphorylation

The NF-κB inducible kinase (NIK) played a crucial role in inducing Nurr1 serine phosphorylation following LPS stimulation, which triggered the association of Nurr1 with the CoREST repressor complex and subsequent gene silencing [42]. To test whether this mechanism also applied to HIV (**Fig 10A**), we knocked down NIK in HC69 cells by using lentiviruses expressing NIK specific shRNAs (**Fig 10B**). The NIK knockdown cells were then induced with a high dose (400 pg/ml) of TNF-α for different time points and a co-IP experiment with a mouse anti-Nurr1 monoclonal antibody and the control IgG was performed (**Fig 10C**). Western blot analysis of the co-IP products using an anti-phospho-serine antibody and an anti-CoREST antibody showed that in cells expressing the control shRNA, the levels of phosphorylated Nurr1 and CoREST pulled down with the anti-Nurr1 antibody increased substantially at 4 hr post-TNF-α stimulation (**Fig 10C**). However, both Nurr1 phosphorylation and its association with CoREST after TNF-α stimulation were largely inhibited in the NIK KD cells.

To examine how NIK KD affects HIV silencing, we performed chase experiments. As shown in **Fig 10D**, GFP expression persisted at 48 hr after TNF-α withdrawal (chase) compared to the control cells. Notably, GFP expression during the chase phase was further decreased in the control cells when they were cultured in the presence of Nurr1 agonist 6-MP. However, the NIK KD cells evidently lost their responsiveness to 6-MP, suggesting that Nurr1 phosphorylation is required for 6-MP mediated HIV silencing as well. Consistent with the GFP expression results, NIK KD also prevented the "shut-down" of Nef expression during the chase phase (**Fig 10E**). These results not only highlight a critical role of NIK in inducing Nurr1 phosphorylation and its association with CoREST for HIV silencing (**Fig 10A**) but also strongly support the trans-repression mechanism for Nurr1-mediated HIV silencing.

## Activation of Nurr1-mediated trans-repression by 6-MP

As a final evidence that Nurr1 mediates HIV silencing by trans-repression, we demonstrated that the Nurr1 agonist, 6-MP, also engaged the CoREST repressor complex. HC69 cells expressing either control shRNA or Nurr1 specific shRNAs were treated with 6-MP for 48 hr and GFP expression was monitored to evaluate HIV transcription during a chase experiment

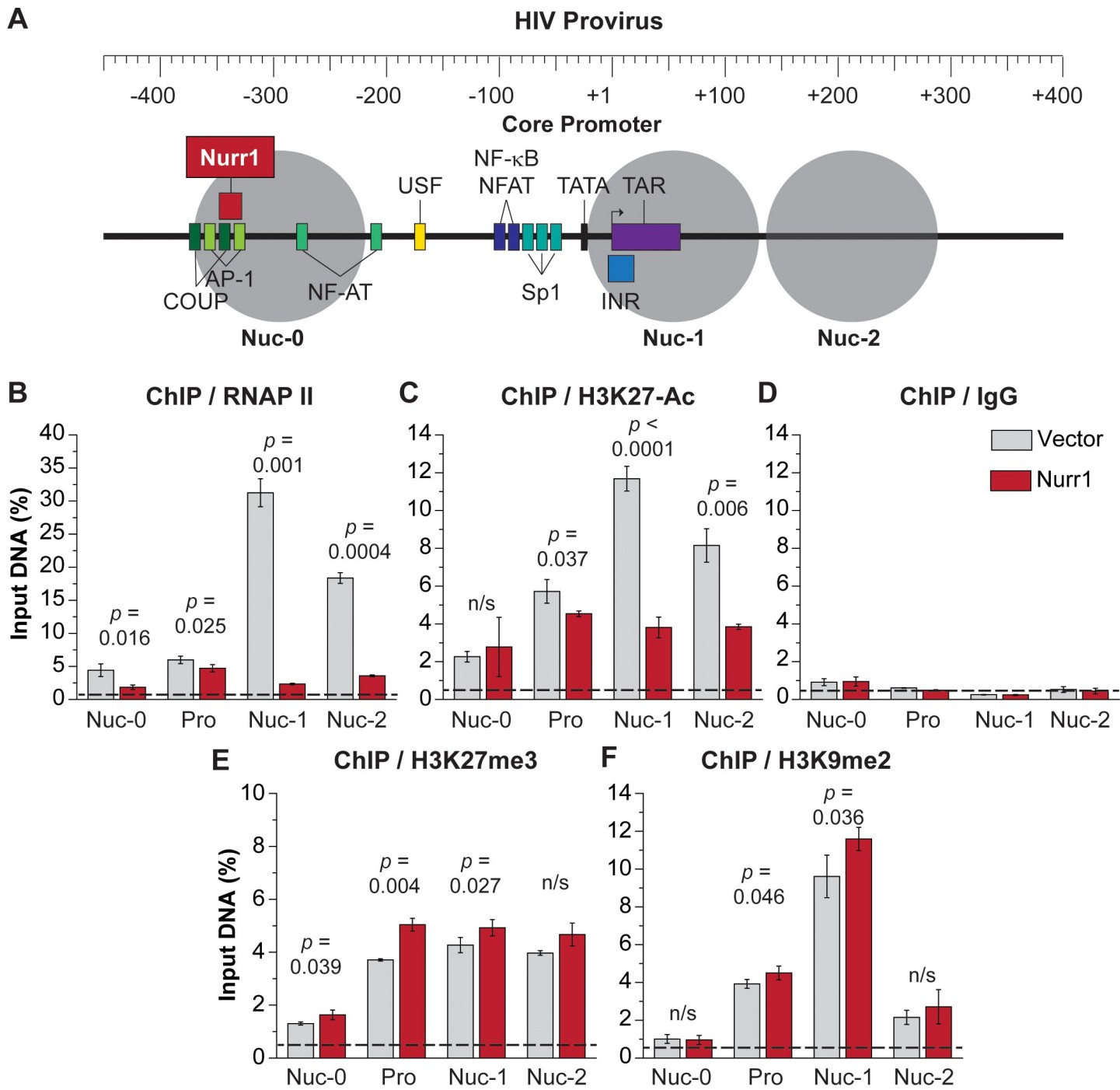

**Fig 8. Nurr1 overexpression promotes repressive histone methylation and deacetylation. A,** Location of Nurr1 binding site and positions of the Nuc0, promoter, Nuc1, and Nuc2 regions in the HIV-1 LTR. **B,** ChIP assays with antibodies for RNA polymerase II (RNAP II). **C,** H3K27-Ac. **D,** control IgG. **E,** H3K27me3. **F,** H3K9me2. HC69 cells infected with either a vector control (grey bars) or a 3X-FLAG-Nurr1 overexpression vector (red bars) were cultured for 48 hr after stimulation with high dose (400 pg/ml) TNF-α for 24 hr. Data is from three technical replicates. Dotted line indicates the IgG background signal.

(S3A Fig). As expected, Nurr1 knockdown increased basal expression approximately 3-fold. In unstimulated control cells, 6-MP treatment reduced the number of GFP+ cells from 30.1% to 15.8% but did not significantly reduce GFP expression in the Nurr1 KD cells. Importantly,

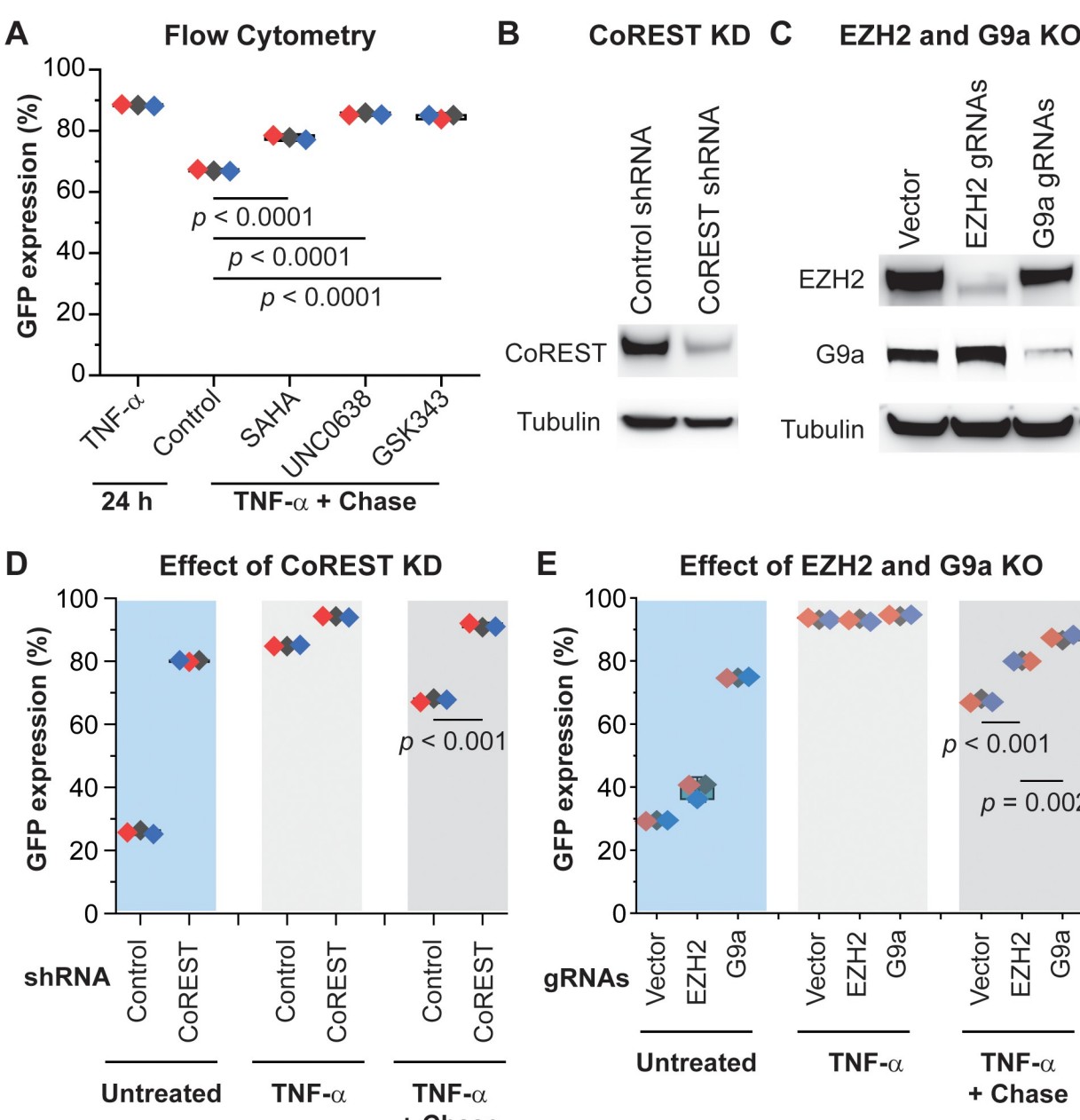

**Fig 9. The CoREST repressor complex plays a pivotal role in silencing active HIV in microglial cells. A,** Inhibition of HDAC1, G9a, and EZH2 blocked silencing of activated HIV in HC69 cells. HC69 cells were stimulated with high dose (400 pg/ml) TNF-α for 24 hr. After washing with PBS, the cells were cultured in the presence of DMSO (placebo, Control), HDAC inhibitor SAHA (1 μM), G9a inhibitor UNC0638 (1.25 μM), or EZH2 inhibitor GSK343 (1.25 μM), for 48 hr. The levels of GFP expression for each treatment were measured by flow cytometry and calculated from three independent experiments, with *p* values between the control and treatment with each inhibitor indicated. **B,** Verification of CoREST KD by Western blot detection of CoREST protein expression in HC69 cell lines stably expressing control shRNA or CoREST-specific shRNA. **C,** Verification of EZH2 and G9a KO by Western blot detection of G9a and EZH2 protein expression in HC69 cells stably expressing CRISPR/Cas9 and G9a or EZH2 specific gRNA, which were compared to the control HC69 cells stably expressing CRISPR/Cas9 without gRNA. β-tubulin was used as a loading control for all Western blot analysis. **D,** CoREST KD prevents HIV silencing. The HC69-control shRNA and HC69-CoREST-shRNA cells were untreated, induced with high dose (400 pg/ml) TNF-α for 24 hr, or used in a chase experiment by continuous culturing the cells for 48 hr after TNF-α stimulation for 24 hr and washes with PBS. GFP expression levels of all cells were measured by flow cytometry and the mean values were calculated from three independent experiments. Significant differences were observed between the HC69-control shRNA and HC69-CoREST shRNA cell lines. **E,** G9a and EZH2 KO prevents HIV silencing. Evaluation of the HC69 cell lines expressing G9a or EZH2 specific gRNA or empty vector by flow cytometry following the same protocol as in panel D. There was a significant difference between HC69-vector and HC69 EZH2 or G9a KO cell lines at 48 hr after TNF-α withdrawal, with *p* < 0.01.

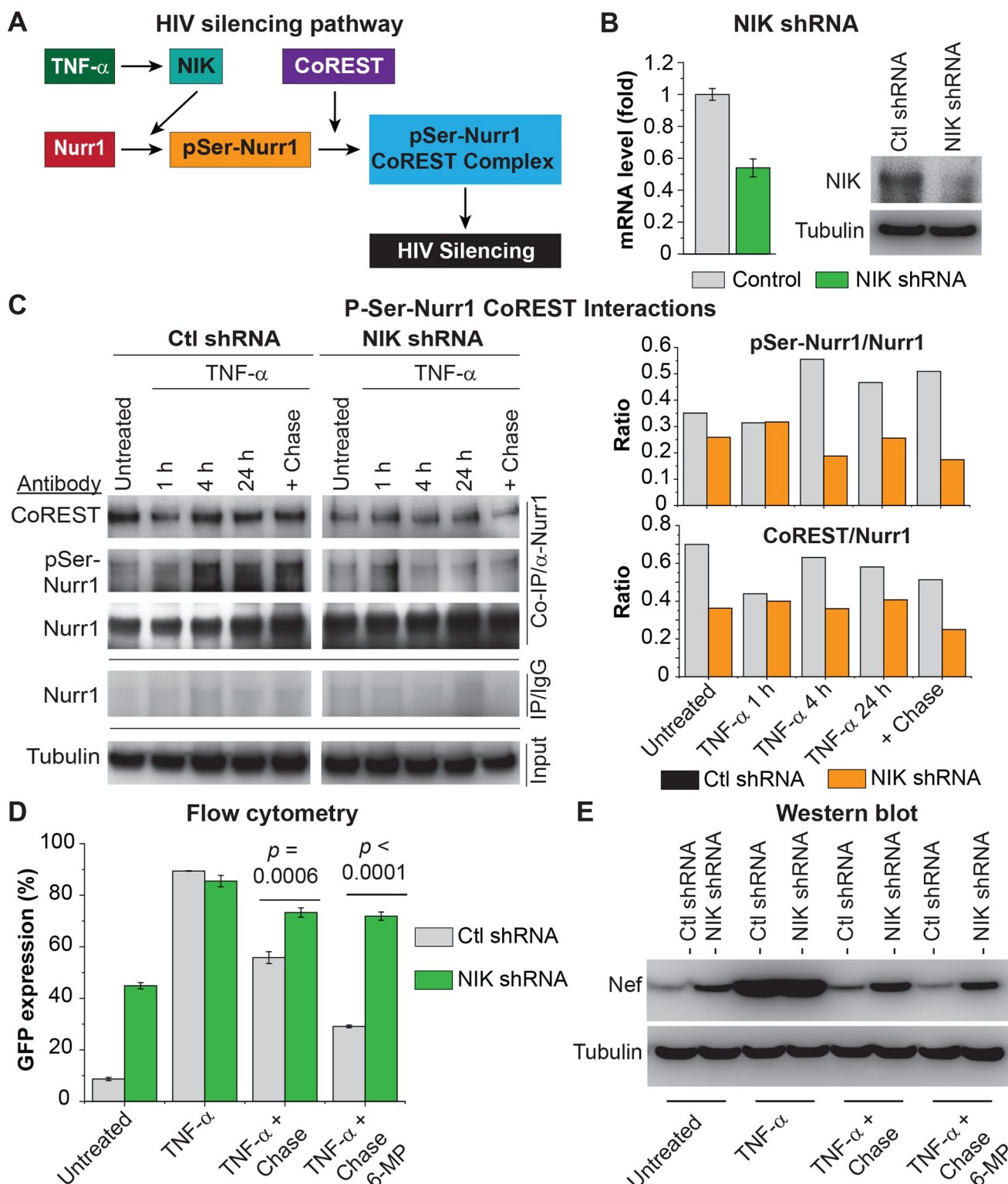

**Fig 10. TNF-α stimulation induces Nurr1 phosphorylation by NIK, which triggers its association with CoREST repressor complex for HIV silencing.** A, schematic presentation of TNF-α induced Nurr1 phosphorylation and its association with CoREST for HIV silencing. B Knock down (KD) of NIK in HC69 cells with NIK specific shRNAs (Santa Cruz Biotech, Cat# sc-36079V) and control shRNA respectively. Left: The relative levels of NIK mRNA from HC69 cells expressing control shRNA (grey bars) and NIK shRNA (green bars) was measured by qRT-PCR using the primers 5' TGCGGAAAGTGGGAGATCCTGAAT 3' (forward) and 5' TGTACTGTTTGGACCCAGCGATGA 3' (reverse). Expression levels were then normalized

to the β-actin gene. Right: Western blot analysis of NIK protein levels using a rabbit monoclonal antibody (abcam Cat # ab203568). C. Co-immunoprecipitation (co-IP) of CoREST, pSer-Nurr1 and Nurr1. Total protein extracts from HC69 cells expressing control shRNA or NIK shRNAs that were untreated, induced with a high dose (400 pg/ml) of TNF-α for different time points, or stimulated for 24 hr followed by a 24 hr chase. co-IP experiments were performed using a mouse monoclonal anti-Nurr1 antibody (Santa Cruz Biotech) or control mouse IgG. Right: The co-IP products were analyzed by Western blot to detect the pulled down Nurr1 protein (using the same anti-Nurr1 antibody), serine-phosphorylated Nurr1 by using a mouse monoclonal anti-phospho-serine antibody from Millipore-Sigma (Cat# P5747-25), and CoREST by using a rabbit polyclonal anti-CoREST antibody (EMD Millipore, Cat# 07–579). Left: Ratios of pSer-Nurr1/Nurr1 and CoRES/Nurr1 in Co-IP samples measured by densitometry of the Western blots. D, GFP expression in HC69 cells expressing either control shRNA (grey bars) and NIK shRNAs (green bars). Cells were untreated, induced with a high dose (400 pg/ml) of TNF-α for 24 hr, or continuously cultured in the absence of TNF-α for 48 hr (chase), in presence or absence of 1 μM 6-MP. The p-values of pair-sample, Student's t-tests for differences in the numbers of GFP+ cells were calculated from three technical replicates. E, Western blot detection of Nef protein in cells treated as described in D. β-Tubulin was used as loading control.

at the end of a chase experiment, the numbers of GFP+ cells in the control cells were reduced from 95% to 46.8% in the absence of 6-MP and were further decreased to 28.8% when cultured in the presence of 6-MP. In contrast, the numbers of GFP+ cells in Nurr1 KD cells were only slightly decreased (from 96.3% to 91.6%) in the absence of 6-MP and remained at 86.6% when cultured in the presence of 6-MP.

To further confirm that 6-MP silences HIV via this mechanism, we stimulated G9a KO HC69 cells and the control cells (**S3B Fig**) with high dose TNF-α for 24 hr, followed by a chase. At the end of the chase period, the number of GFP+ cells in the vector infected control cells went down from 91.7% to 54% in the absence of 6-MP and further decreased to 31.7% in the presence of 6-MP. In contrast, the numbers of GFP+ cells in G9a KO cells only slightly decreased from 91.9% to 90.4% in the absence of 6-MP and remained at 86.6% in the presence of 6-MP.

We next conducted ChIP assays to determine if 6-MP treatment enhances recruitment of the CoREST repressor complex to HIV 5'LTR. As expected, 6-MP treatment dramatically increased the levels of CoREST and G9a at the HIV LTR but only had a modest effect on HDAC1 levels (**S3C Fig**).

Finally, to confirm that Nurr1 is also critical for the silencing of HIV in primary microglial cells, we infected iPSC-derived human microglial cells (iMG) with the same HIV reporter virus described earlier (**Fig 1A**). About 50% of the iMG became GFP+ two days after HIV infection (**Fig 11A**). We then treated the infected iMG with 6-MP and another Nurr1 agonist, amodiaquine (AQ) [48,55], for four days. Both 6-MP and AQ decreased the number of GFP + cells in a dose-dependent manner (**Fig 11B** and **11C**) and lowered the levels of HIV un-spliced transcripts (**Fig 11D**). Both agonists also dose-dependently reduced MMP2 mRNA in iMG (**Fig 11E**). Collectively, results from both hμglia and iMG strongly suggested an important role for Nurr1 in HIV silencing in microglial cells.

Thus, our results clearly demonstrate that 6-MP silences HIV through activation of Nurr1 and fortifies our hypothesis that this Nurr1 agonist silences HIV by engaging the CoREST repressor complex.

## Nurr1 drives activated microglial cells towards homeostasis

Our RNA-Seq data also provided important insights into the cellular pathways that were impacted by Nurr-1 over- and under-expression. We focused our attention on the changes in cellular transcriptome during the chase step following TNF-α induction since, as described above, this is the stage where Nurr1 has the greatest impact on HIV gene expression. The volcano plots in **Fig 12A** show that a disproportionate number of genes are downregulated in Nurr1 overexpressing cells. This is also illustrated by the differential gene expression curves in **S4A Fig,** showing that a small subset of genes are selectively up and down regulated during the chase. The heat maps in **S5A and S5B Fig** further illustrate this point by showing the maximally differentially regulated genes in cells where Nurr1 is either overexpressed or knocked

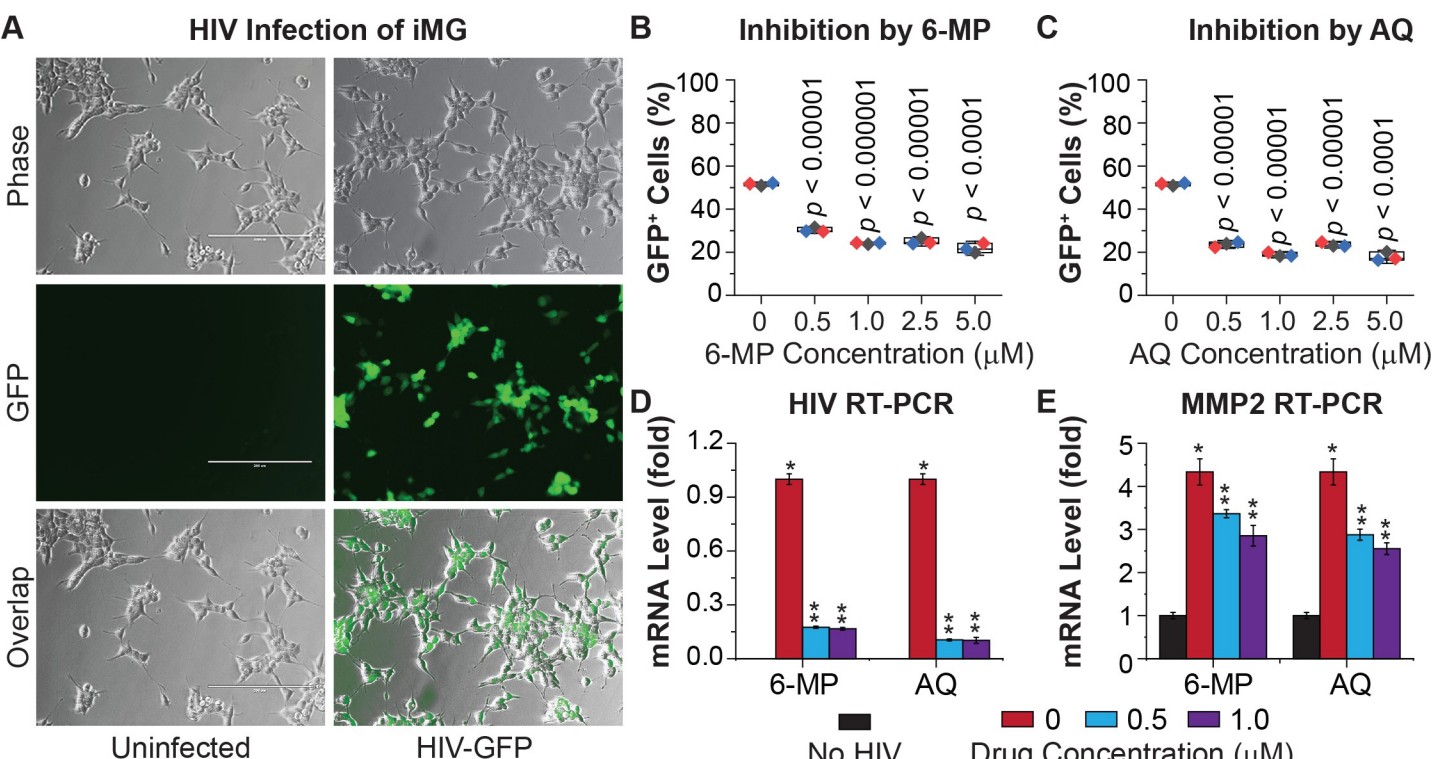

**Fig 11. Nurr1 Mediates HIV silencing in iPSC-derived microglial cells (iMG). A,** Representative phase contrast, GFP, and overlapped images of iMG that were un-infected or infected with the reporter HIV-1 shown in Fig 1A, at 48 hr post-infection (hpi). HIV-infected iMG were treated with different doses of Nurr1 agonist 6-MP or AQ for four days, followed by flow cytometry analysis of GFP expression. **B,** The average levels of GFP expression in iMG treated with various doses of 6-MP were calculated from three replicates. **C,** The average levels of GFP expression in iMG treated with various doses of AQ were calculated from three replicates. **D,** The average levels of HIV RNA (un-spliced) in the cells described in panels **A** and **B**, were measured by RT-qPCR and calculated from three replicates of qRT-PCR. The mean value of HIV transcript from HIV infected but untreated cells was referred to as level "1". **E,** The mRNA level of Nurr1 target gene MMP2 in the same cells described in **D** was measured by qRT-PCR. The mean value of MMP2 mRNA from three replicates of un-infected cells was referred to as level "1". The average levels of HIV transcript and MMP2 mRNA in each sample were calculated from three technical replicates. Differences in HIV and MMP2 mRNA levels between untreated cells and cells treated with different doses of 6-MP or AQ were statistically significant (** $p$-values $<0.001$). HIV transcripts were only detected in infected iMG cells (panel **D**). MMP2 mRNA was significantly elevated in HIV infected iMG (panel **E**).

down at each stage of the chase experiment. Pathways that showed the most statistically significant changes in response to Nurr1 overexpression included the downregulation of key pathways with critical roles in cellular proliferation and metabolism including: MYC, E2F and MTORC signaling and G2M checkpoint (S4B Fig). By contrast, KD of Nurr1 by shRNA did not selectively activate any major signaling pathways. It is important to note that Nurr1 over-expression did not significantly interfere with the TNF-α signaling pathway during any step of these experiments, suggesting that the cellular proliferation pathways we have identified are directly regulated by Nurr1.

To further address this issue and determine whether Nurr1 simply accelerated the reversal of the normal microglial response to TNF-α stimulation during the chase, or if it regulated a distinct set of genes and pathways, we performed a gene trajectory analysis (S6 to S8 Figs). The pseudo-trajectory was defined as containing three steps: Step 1 defines the changes in gene expression following stimulation with a low dose of TNF-α (20 pg/ml, to simulate a sub-optimal activation signal) compared to untreated cells. Step 2 defines additional changes after stimulation with high dose TNF-α compared to cells treated with low dose TNF-α. Step 3 defines the gene expression changes following the chase step compared to cells treated with high dose TNF-α. For each of these steps we calculated whether the expressed protein-coding

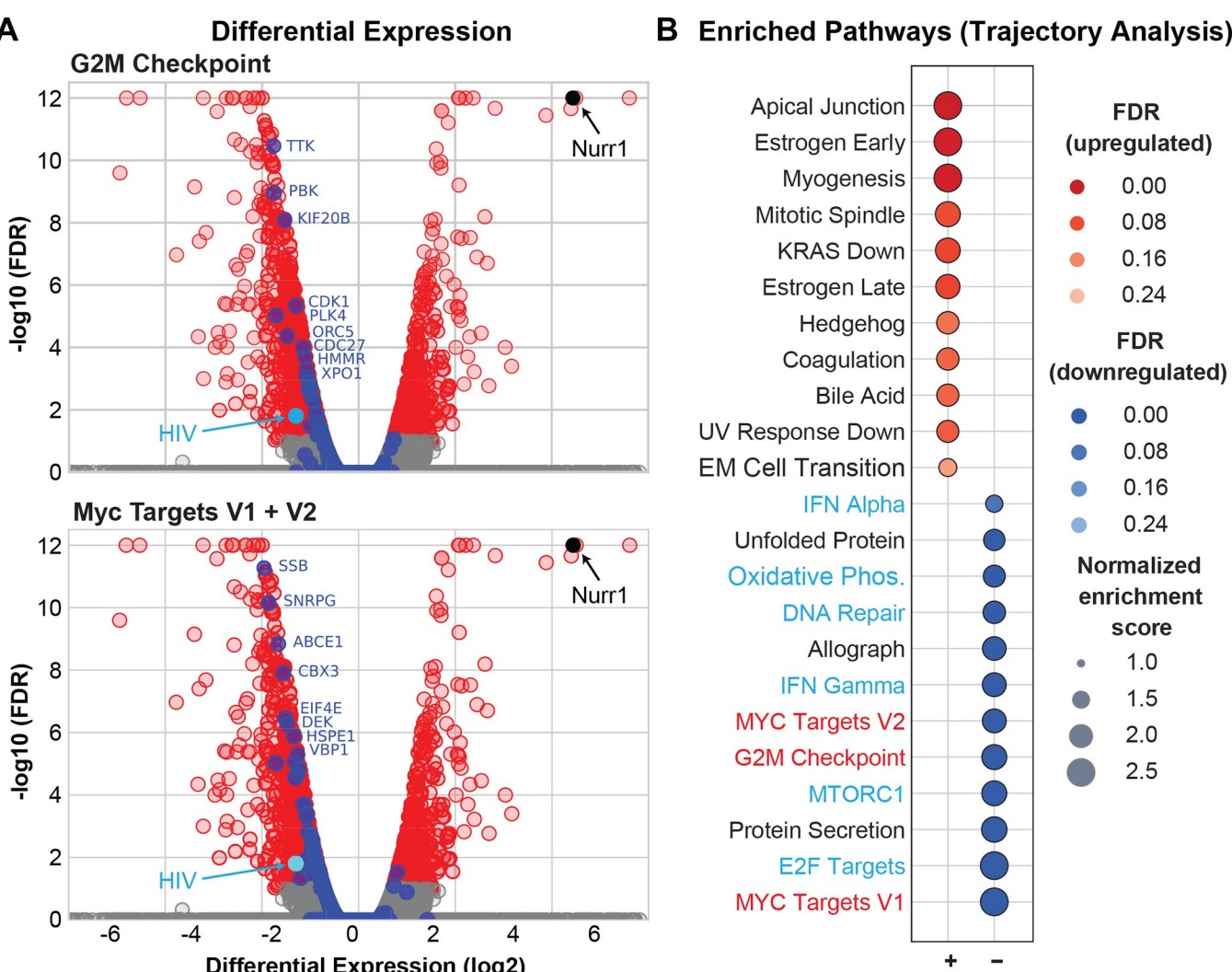

**Fig 12. Identification of genes that are silenced due to Nurr1 overexpression. A.** Volcano plots showing differential expressed genes during the chase step in Nurr1 overexpressing cells. The position of Nurr1 is shown in black. HIV is shown in light blue. Genes from the G2M Checkpoint (top) and Myc Target (Bottom) pathways are heighted in dark blue. Statistically non-significant genes are shown in grey. **B.** Pathway analysis of genes differentially expressed in Nurr1 overexpressing cells. The genes were selected based on Trajectory analyses. Pathways that are up regulated are shown in red. The color and size of circles correspond to statistical significance, as shown by FDR, and normalized enrichment values, respectively. Positive (red) and negatively (blue) enriched pathways are shown in the left and right plot, respectively.

genes were either upregulated (designated as "u"), downregulated (designated as "d") or did not show differential expression in a statistically significant manner (designated as "n"). Most genes did not show any change in their expression following the above treatments (designated as the "nnn" group) in both control (vector) and Nurr1-overexpressing cells (**S6A Fig**). As expected, control cells had higher numbers of nnn group genes than Nurr1 overexpressing cells. Among those genes that showed an expression change in Nurr1 overexpressing cells, the majority exclusively changed their expression profiles during the chase step (i.e., nnu and nnd trajectories, **S7 Fig**) and were not differentially expressed in response to TNF-α treatment.

To further characterize the Nurr1-specific changes in expression patterns, we used the list of genes in each of the trajectories identified in Nurr1 overexpressing cells and defined their trajectory in control cells (**S8 Fig**). This analysis showed that over 1400 and ~800 of the genes

that fall into the nnd or nnu trajectories in Nurr1 overexpressing cells, respectively, have the nnn trajectory in control cells (**S8 Fig**). Thus, the main transcriptomic outcome of Nurr1 over-expression compared to control cells is the induction of changes in expression of a group of genes exclusively during the chase step.

We used this subset of genes to define the functional impact of Nurr1 using pathway analyses (**Figs 12B** and **S6B**). Strikingly, the Nurr1-induced changes were robustly characterized by the downregulation of several key proliferative pathways, including: MYC, E2F and MTORC signaling, G2M checkpoint regulation, metabolic pathways such as oxidative phosphorylation, and inflammatory pathways such as IFN-α and IFN-γ response pathways (**S6B Fig**). By way of illustration, genes from the G2M and Myc Target groups are highlighted on the volcano plots in **Fig 12A**.

Heat maps of the differentially expressed genes further emphasized that the vast majority of genes in each pathway were downregulated in Nurr1 overexpressing cells. For example, among 69 and 60 represented MYC and E2F target genes, 66 and 54 were downregulated in Nurr1 overexpressing cells, respectively (**S9 Fig**). Another compelling way of visualizing these results is to apply the pattern of expression of the Nurr1-specific genes to the KEGG cell cycle pathway (**S10 Fig**). The strong downregulation by Nurr1 at multiple steps in the cell cycle control pathway is immediately obvious.

As expected, the differentially expressed genes, were highly enriched for NF-κB motifs in their promoters, consistent with TNF-α activation for genes that were down-regulated by Nurr1 during the chase (**S11 Fig**). Similarly with the subset of regulated genes that had one or more Nurr1 binding sites were also enriched in genes subject to NF-κB regulation (**S12A Fig**). Differential expression plots (**S12B Fig**) showed that the majority of genes with Nurr1 binding sites were actually upregulated in cells that over-express Nurr1. However, an important subset of genes, including HIV are down-regulated in untreated cells and the proportion of these genes increases markedly during the chase. As described previously, HIV expression is not enhanced by Nurr1 expression in response to the initial TNF-α activation.

Thus, the main impact of Nurr1 on the overall cellular response to inflammatory cytokines, in this case TNF-α, was to accelerate the cellular return to homeostasis by shutting down pathways involved in inflammation and microglial activation. While these anti-inflammatory, pro-homeostasis effects could indirectly lead to HIV proviral transcriptional shutdown, the enhanced downregulation of HIV expression in Nurr1 overexpressing cells, even under basal untreated conditions, suggests that in addition to its pro-homeostasis effects, Nurr1 also directly regulates the expression of the HIV provirus.

## Discussion

### HIV latency in microglial cells

Microglial cells are one of the major cellular reservoirs of HIV in the CNS [13,14]. These long-lived cells contribute to increased neuroinflammation and oxidative stress [13,71], and development of HAND by secreting a variety of neurotoxins as well as harmful HIV proteins such as gp120, Tat, and Rev [72,73]. Eradication or complete silencing of HIV-infected microglial cells is therefore crucial not only for an HIV cure, but also to prevent the development of HAND, which affects the majority of HIV infected individuals.

Previous studies involving HIV-1 infection of transformed cell lines suggested that epigenetic regulation plays a major role in the establishment and persistence of HIV latency in astrocytes and microglial cells [74,75]. The cellular COUP transcription factor (COUP-TF) interacting protein (CTIP2) forms a large transcriptional repressor complex with epigenetic silences including the histone deacetylases HDAC1/2, the histone methyltransferases

SUV39H1 and SET1, the lysine(K)-specific demethylase KDM1, and heterochromatin protein1 (HP1) [76,77]. Recruitment of this complex to HIV-1 promoter leads to proviral genome silencing due to reduced histone acetylation and increased levels of histone 3 tri-methylations at lysine 9 (H3K9me3) [25,76–78]. At the same time, CTIP2 forms another complex with CDK9, Cyclin T1, HEXIM1, 7SK snRNA, and high mobility group AT-hook 1 (HMGA1), which is also recruited to HIV-1 promoter [77,78]. In the absence of HIV-1 Tat, this complex with inactive pTEFb further supports HIV-1 latency by preventing elongation of RNA polymerase II for active transcription [23]. Nevertheless, it remains unknown if these mechanisms also apply to HIV-infected primary microglial cells as transformed cells often behave quite differently.

## Silencing of HIV by Nurr1 and CoREST

Nuclear receptors are special transcription factors that turn on or turn off expression of target genes upon specific ligand binding [79]. For example, the estrogen receptor (ER) has been found to promote HIV latency in T cells [80]. In our previous work, we demonstrated that autocrine inflammatory cytokines such as TNF-α were major drivers for spontaneous HIV reactivation in microglial cells [18]. Activation of the glucocorticoid receptor (GR) with its specific ligands such as dexamethasone antagonized the effects of cytokines on HIV reactivation [20]. However, we observed that the reactivated HIV was subsequently silenced in microglial cells even in the absence of dexamethasone, suggesting the existence of additional HIV silencing mechanisms.

In the present study, we identified the nuclear receptor Nurr1 as a key HIV silencing factor. Overexpression of Nurr1 had little effect on preventing reactivation of latent HIV but strongly enhanced silencing of active HIV after TNF-α stimulation and subsequent withdrawal. Conversely, KD of endogenous Nurr1 in HC69 cells inhibited silencing of active HIV after TNF-α withdrawal. Thus, results from both overexpression and KD experiments unequivocally demonstrated a pivotal role of Nurr1 in silencing active HIV. A role for Nurr1 in HIV silencing was observed in multiple clones of immortalized human microglial cells and in iPSC-derived human microglial cells. The repression of HIV transcription is independent of the initial activator and can be readily observed during chases of cells stimulated by LPS, poly(I:C) and TNF-α.

In contrast, we did not see significant effects of the related receptors, Nur77 and Nor1 on the repression of HIV after reactivation. However, since it is well known that the different Nerve Growth Factor IB-like nuclear receptors interact with each other or with other nuclear receptors such as GR and RXR [81,82], Nur77 and Nor1 may contribute to the control of HIV expression in other contexts.

Nurr1 binds directly to the HIV provirus at a site in the U3 region of the LTR which overlaps the previously mapped COUP/AP1 site [83,84], which has also been identified as a PPAR response element [85]. Nurr1 binding directly represses HIV transcription during latency and when microglial cells are exposed to low doses of TNF-α and other pro-inflammatory cytokines. However, this block is insufficient to restrict HIV expression in response to a high dose of TNF-α.

Nurr1 recruits the CoREST/HDAC1/G9a/EZH2 repressor complex to the HIV LTR in a mechanism analogous to repression of cellular early response genes [42]. Specifically, the NF-κB inducible kinase (NIK) promotes serine phosphorylation of Nurr1, which triggers its association with and subsequent recruitment of the CoREST transcription repressor complexes to the HIV promoter.

CoREST, in turn, serves as a scaffold to recruit HDAC 1/2 and other epigenetic silencers such as the histone lysine methyltransferases (HKMTs) G9a and EZH2. Functional inhibition

of the HKMTs with small molecule inhibitors, or KD or KO of each component of the repressor complex including CoREST, G9a, and EZH2 strongly inhibited HIV silencing. In addition to repressing the HIV LTR, data from RNA-Seq analysis indicated that Nurr1 directs epigenetic silencing to an important subset of cellular genes that regulate microglial cell activation.

## Anti-inflammatory role for Nurr1 in HIV-infected microglial cells

Over the past decade, the intimate relationship between neuroinflammation, neurodegeneration and abnormal activation of microglial cells has been implicated in a wide range of diverse neurological diseases [86–91]. The accumulation of inflammatory cytokines and proteases in the substantia nigra and the striatum, as well as activation of the microglia [92], are also common features of HAND [13,93].

There are now compelling reasons to believe that in addition to providing reservoirs for HIV, the abnormal physiology of HIV-infected microglia also plays a critical role in the development of HAND. Microglia constitute the first barrier of the innate immune response in the brain and become activated and polarized to maintain the integrity of the CNS [94,95]. Microglial activation is not only triggered by inflammatory cytokines and neurochemical mediators [96–99], but activated microglia themselves secrete exaggerated amounts of neurotoxins, including TNF-α, nitric oxide, IL-6, IL-1β, reactive free radicals, and matrix metallopeptidases (MMPs) [100,101]. Since the production of these pro-inflammatory factors by microglia is augmented by HIV infection, we hypothesize that autocrine signaling leads HIV infected microglial cells to undergo cycles of spontaneous reactivation [19–21,102] and subsequent silencing by Nurr1.

Nurr1 also protects dopaminergic neurons from inflammation-induced neurotoxicity through the inhibition of pro-inflammatory cytokine expression in microglia and astrocytes by recruiting CoREST corepressor complexes to NF-κB target genes [42,103]. Multiple studies have reported that activation of Nurr1 reduces inflammation, protects neurons, and decreases Parkinson's disease related symptoms [45,53,55,104]. Conversely, Nurr1 deficiency or reduced expression due to mutations in adults is also a major contributing factor in the pathogenesis of Parkinson's disease [105,106]. Although a reduction of Nurr1 expression in neurons does not directly affect their survival, the survival rate of neurons decreases in response to inflammatory stimuli when Nurr1 is deficient [42].

Data from our RNA-Seq experiments confirms that Nurr1 overexpression directs activated microglial cells towards homeostasis by repressing NF-κB signaling pathway, metabolism, cell cycle control, MYC signaling and IFN-α and INF-γ responses. This finding is consistent with a recent report that glycolysis downregulation is a hallmark of HIV-1 latency in microglial cells [107]. Thus, in addition to silencing HIV, Nurr1 apparently plays a crucial role in suppression of microglia activation. It seems likely that similar Nurr1 dependent silencing mechanisms also exist in perivascular macrophages, which express Nurr1 and can harbor HIV-1 in the CNS.

Further studies are warranted to determine whether enhancing Nurr1 and related receptors activity will be beneficial for HIV patients with HAND. Multiple Nurr1 agonists exhibit strong therapeutic effects and potentials for Parkinson's disease in pre-clinical animal study and human trials [108]. Similarly, treatment of mice with isoxazolo-pyridinone 7e, an activator of Nurr1 signaling pathway, reduced the incidence and the severity of disease in an experimental model of autoimmune encephalomyelitis (EAE) [109]. In this study, we tested the Nurr1 agonists 6-MP and AQ. Both agents strongly inhibited expression of HIV and the neurotoxin MMP2 in HC69 cells and iMGs. It would therefore be of great interest to test additional agonists, particularly those new generations of Nurr1 agonists currently on pre-clinical and

human trials, for their anti-HIV activity and eventual application in the clinic for treatment of HAND.

## Materials & methods

### Chemicals and reagents

TNF-α (Invitrogen, Cat. #PHC3015) was used to induce HIV-1 reactivation in microglial cells. Nor1 and Nurr1 agonists 6-mercaptopurine (6-MP) (Millipore-Sigma, Cat#38171) and amodiaquine (AQ) (Millipore-Sigma, Cat#SMB00947) were used to activate the Nerve Growth Factor IB-like nuclear receptors. GSK343 (Sigma Aldrich, Cat# SML0766), UNC0638 (Sigma Aldrich, Cat#U4885), and suberoylanilide hydroxamic acid (SAHA, Millipore-Sigma, Cat#SML0061) were used to examine the effects of EZH2, H9a, and HDAC1/2 on HIV silencing respectively.

### Cells and flow cytometry analysis of HIV/GFP expression

HIV-1 infected immortalized human microglial (hμglia) HC69 cells were cultured and maintained as described as previously [18]. Induced pluripotent stem cells (iPSC)-derived human microglial cells (iMG) (Tempo Bioscience, Cat#SKU 1001.1) were plated, allowed to differentiate and maintained in culture on plates pre-coated with Matrigel matrix (Corning, Cat#356254) according to the manufacturer's instructions. The iMG were infected with EFGP HIV-1 reporter virus at 1 to 1 (cell-to-virus moiety), which was produced, purified, and titrated as described previously [18]. Two days after infection, the iMG were treated with and without the Nurr1 agonists 6-MP and AQ for four days. Infected with the same EGFP-reporter HIV-1 virus (**Fig 1A**), HIV expression in hμglia and iMG cells was measured and quantified with percentage (%) of GFP+ cells by flow cytometry as described previously [20,21].

### Lenti-viral construction and production, and generation of stable cell lines

Three lentiviral constructs, pLV[Bxp]-Bsd-CMV>3xFLAG-Nur77, pLV[Bxp]-Bsd-CMV>3xFLAG-Nurr1, and pLV[Bxp]-Bsd-CMV>3xFLAG-Nor1 were generated by inserting the full-length open reading frame (ORF) of human NR4A1 (Nur77), NR4A2 (Nurr1), and NR4A3 (Nor1) cDNA fragment into the empty vector pLV[Bxp]-Bsd-CMV>3xFLA immediately downstream of the Kozak sequence (VectorBuilder, vector ID: VB180227-1135bmn, VB180227-1134jht, and VB180227-1136rwc). The inserted cDNA was also "in frame" fused with the coding sequence of the N-terminal 3X-FLAG peptide tag, allowing to generate N-terminal 3xFLAG-tagged proteins. A fusion PCR strategy was used to generate the 3X-FLAG-Nurr1 C280AE281A [CEAA] mutant. The first PCR fragment was generated by using pLV [Bxp]-Bsd-CMV>3xFLAG-Nurr1 as template with the forward primer 5'CGGTAGGCGTG TACGGTGGGAGGTC3' (derived from vector sequence located before the Kozak sequence) and the reverse primer 5'CTTTGCAGCCCGCAGCGGTGCGCACGC3' (derived from the Nurr1 C280E281 region, with C280 codon (TGT) converted to A280 (GCT) and E281 codon (GAG) converted to A281 (GCG). The second PCR fragment was also generated from the same template DNA with the forward primer 5'GCGTGCGCACCGCTGCGGGCTGCA AAG3' (derived from the C280E281 region and with C280E281 converted to A280A281) and the reverse primer 5'TGATAGTCAGGGTTCGCGTGGAACC3' (derived from Nurr1 coding sequence downstream of the C280A281 region). The two PCR fragments, which overlap each other at the C280AE281A mutation region, were used as templates for a fusion PCR by using the forward primer of the PCR fragment and the reverse primer of the second PCR fragment. The resulting fusion PCR fragment containing the expected mutations and an unique Cla1

restriction site at its 5' end and a Sal1 restriction site at its 3'end, was swapped into pLV[Bxp]-Bsd-CMV>3xFLAG-Nurr1 after Cla1 and Sal1 double digestion and subsequent ligation. Successful mutation and integrity of the resulting lentiviral plasmid was confirmed by DNA sequencing. **Two lentiviral constructs expressing human Nurr1-specific shRNA (5'GG TTCGCACAGACAGTTTAAA3' and 5'ATACGTGTGTTTAGCAAATAA3'), one lentiviral construct expressing human CoREST-specific shRNA (5'CCCAATAATGGCCAGA ATAAA3'), and two lentiviral constructs expressing control shRNAs (5'**CCTAAGGTTA AGTCGCCCTCG3' and 5' CAACAAGATGAAGAGCACCA3') were purchased from VectorBuilder. All lentiviral constructs carried an ampicillin resistance gene for selection in bacteria (*E. coli*) and a blasticidin resistance gene for selection of stable expression in mammalian cells. **Infectious viral particles with each of these lentiviral constructs were produced by cotransfecting 293T cells with** packaging plasmid psPAX2 (Addgene, Cat#12260) and Env Vector pCMV-VSVg (Addgene, Cat#138479)**. HC69 cells stably expressing 3X-FLAG-Nur77, 3X-FLAG-Nurr1, 3X-FLAG-Nor1, empty vector, gene-specific shRNA and control shRNA were generated by infection of the cells with purified lentiviral particles for two days, followed by culturing the cells in the presence of blasticidin at 10 μg/ml.**

To investigate the effects of G9a and EZH2 on HIV silencing, we conducted CRISPR/Cas9 mediated "knocking out" (KO) of these genes in HC69 cells, using a dual CRISPR/Cas9 gRNA lentiviral vector. Two different guide RNAs targeting EZH2 (TGAGCTCATTGCGCGGGACT and GATCTGGAGGATCACCGAGA) or G9a (TTCCCCATGCCCTCGCATCC and GTG GCAGCCCCACGGCTGAA) were cloned into lentiCRISPR v2-Blast plasmid following the protocol described previously [110]. LentiCRISPR v2-Blast was a gift from Mohan Babu (Addgene plasmid # 83480). VSV-G pseudotyped viruses expressing CRISPR/Cas9 gRNAs were produced in HEK 293T cells by transfection of lentiCRISPR v2 plasmids together with psPAX2 and pCMV-VSV-G. HC69 cells infected with the EZH2 or G9a KO lentiviruses were cultured in the presence of blasticidin (10 **μg/ml).** Successful KO of these genes in HC69 cells were verified by Western blot analysis of EZH2 and G9a proteins in the resulting cell lines.

## Reverse transcription and quantitative polymerase chain reaction (RT-qPCR)

Total RNAs from HC69 or HIV-infected IMG cells with different treatments were isolated by using the RNeasy Plus Mini kit from Qiagen (Cat#74134). The purified total RNAs were converted to first-strand cDNAs by using a reverse transcription kit (Bio-Rad, Cat#1708891). The relative levels of HIV-1 un-spliced transcript and human MMP-2 mRNA were measured by qRT-PCR using the primers 5'GGGTGCGAGAGCGTCGGTATTAAGC3' (HIV-1 un-spliced-forward) and 5'TCCTGTCTGAAGGGATGGTTGTAGC3' (HIV-1 un-spliced-reverse), and 5'ATAACCTGGATGCCGTCGT-3′ (MMP2 forward) and AGGCACCCTT-GAAGAAGTAGC-3′ (MMP2 reverse), respectively. The mRNA level of the housekeeping gene β-actin in each sample was used as reference for normalization, which was measured by qRT-PCR using the primers 5'-TCCTCTCCCAAGTCCACACAGG-3′ (forward) and 5'-GGGCACGAAGGCTCATCATTC-3′ (reverse). Each qRT-PCR was conducted in triplicates.

## ChIP and ChIP-seq analyses

Standard procedures were followed for all ChIP assays. Briefly, cells were fixed with 1% Formaldehyde for 10 minutes (min) at room temperature, followed by incubation in PBS containing 125 mM glysine for 10 min at room temperature. After two washes with ice-cold PBS, cells were re-suspended and allowed to swell in CE buffer [10 mM HEPES, pH7.9, 60 mM KCl, 1 mM EDTA, 0.5% NP-40, 1 mM DTT] on ice for 10 min. After centrifugation at 2,000 g for 10

min at 4˚C, nuclei were re-suspended in SDS lysis buffer [50 mM Tris-HCl, 1 mM EDTA, 0.5% SDS] and incubated on ice for 10 min. Sheared chromatins were prepared by sonicating the nuclei lysate to generate DNA fragments in the range of 250 to 500 bps. ChIP assays with specific antibodies were carried out in ChIP dilution buffer [16.7 mM Tris-HCl, pH 8.1, 167 mM NaCl, 1.2 mM EDTA, 1.1% Triton X-100, and 0.01% SDS] containing 5 μg antibody and 50 ul protein-A/protein-G magnetic beads per reaction at 4˚C for overnight with rotation, followed by consecutive washes with low salt wash buffer [20mM Tris-HCl, pH8.1, 150 mM NaCl, 1 mM EDTA, 1% Triton X-100, 0.1% SDS], high salt wash buffer [20mM Tris-HCl, pH8.1, 500 mM NaCl, 1 mM EDTA, 1% Triton X-100, 0.1% SDS], and RIPA buffer [20 mM Tris-HCl, pH7.5, 150 mM NaCl, 5 mM EDTA, 0.5% Triton X-100, 0.5% sodium deoxycholate, and 0.1% SDS]. The washed beads were then re-suspended in elution buffer [50 mM Tris-HCl, pH 6.5, 20 mM NaCl, 100 mM NaHCO3, 1 mM EDTA, 1% SDS, 100 μg/ml proteinase K] and incubated at 50˚C for 2h. Supernatants from the beads were collected and used for ChIP DNA purification using Qiagen's PCR purification kit (Cat#28104). Quantification of input and ChIP DNA corresponding to HIV-1 promoter region was conducted by qPCR using specific primers as reported previously [111].

ChIP assays were conducted by using antibodies to H3K27me3 (Millipore, Cat# CS200603), H3K9me2 (Cell Signaling, Cat# 4658), H3K27-Ac (Cell Signaling, Cat# 8173S), RNA polymerase II (Millipore, Cat# CS200572), and control IgG, respectively. ChIP products and input DNA were quantified by qPCR by using the following primers:

1. Nuc-0: 5' ACACACAAGGCTACTTCCCTGA 3' (-390F) and 5' TCTACCTTATCTGGCT CAACTGGT 3'(-283R);

2. HIV promoter: 5' AGCTTGCTACAAGGGACTTTCC 3'(-116F) and 5' ACCCAGTACA GGCAAAAAGCAG 3'(+4R);

3. Nuc-1: 5' CTGGGAGCTCTCTGGCTAACTA 3'(+30F) and 5' TTACCAGAGTCACACA ACAGACG 3'(+134R); and

4. Nuc-2: 5' GACTGGTGAGTACGCCAAAAAT 3'(+283F) and 5' TTTCCCATCGCGATCT AATTC 3' (+390R).

For ChIP-Seq analyses, the DNA products from each ChIP assay were first end repaired with end repair enzyme mix (New England Biolabs, Inc., Cat#M6630), then ligated to NEB-Next adaptor included in the NEBNext Ultra II DNA Library Prep Kit for Illumina (Cat#E7645L) according to the manufacturer's instruction, followed by PCR amplification with a specific pair of bar-coded primers. Next, to enrich HIV-1 specific sequences in the library, DNA samples from all ChIP assays were pooled, denatured at 98˚C for 10 min, and then subjected to hybridization with 50 times excessive amount of biotin-labelled and pre-denatured HIV-1 genomic DNA in hybridization buffer containing 5XSSC and salmon sperm DNA (100 μg/ml) at 65˚C for 1 h. Fragments hybridizing to biotin-labelled HIV-1 DNA were pulled down by incubating the hybridization reaction with streptavidin-conjugated magnetic beads (ThermoFisher Scientific, Cat#88816) at room temperature for 30 min, followed by three times washes with ion wash buffer and elution in water. The enriched ChIP library DNA was PCR amplified with Ion A and Ion P1 primers, and PCR fragments in the range from 300 to 500 bps were purified from agarose gel after electrophoresis and loaded for Ion Torrent sequencing.

We aligned the sequence reads to NL4.3-Cd8a-EGFP-Nef+ HIV-1 genome. Raw fastq sequencing data were imported to *the public server at* usegalaxy.org *for analysis* [112]. *We used* FASTX-Toolkit for deconvolution of reads. Read mapping was performed by Bowtie2 tool

with default settings using the NL4.3-Cd8a-EGFP-Nef+ HIV-1 as a reference genome [113,114]. DeepTool2 was used to make graphs for distribution of mapped reads along HIV-1 genome [115].

## RNA-Seq and data analysis

Approximately 2 million GFP-negative cells from each of the cell lines HC69-3X-FLAG-vector, HC69-3X-FLAG-Nor1, HC69-3X-FLAG-Nurr1, HC69-control shRNA, and HC69-Nurr1 shRNA were collected from sorting. The isolated cells were expanded in DMEM culture media with low glucose (1g/L) and 1% FBS for 48 hr in the presence of dexamethasone (1μg/ml) to maintain HIV latency as reported previously [20]. The cells were next cultured in fresh medium without dexamethasone, un-treated, or treated with low dose (20 pg/ml) and high dose (400 pg/ml) TNF-α for 24 hr. One portion of the cells treated with high dose TNF-α were washed twice with PBS, followed by culturing in fresh medium in the absence of TNF-α and dexamethasone for 48 hr. Total RNAs from each cell line with different treatments were isolated by using the RNeasy Plus Mini kit from Qiagen (Cat#74134). The isolated RNAs were treated with RNase-free DNase I at 37˚C for 30 min to remove genomic DNA, followed by a second-round purification using the same RNA purification kit. For reproducibility concerns, the RNA-Seq analysis consisted of RNA samples from two independent experiments performed several months apart. The RNA-seq data has been deposited in the NCBI Sequence Read Archive (SRA) and is available under BioProject accession PRJNA789419.

Total cellular RNA was subjected to 150 base long, paired end RNA-Seq on an NovaSeq 6000 instrument. RNA-Seq reads were quality controlled using Fastqc and trimmed for any leftover adaptor-derived sequences, and sequences with Phred score less than 30 with Trim Galore, which is a wrapper based on Cutadapt and FastQC. Any reads shorter than 40 nucleotides after the trimming was not used in alignment. The pre-processed reads were aligned to the human genome (hg38/GRCh38) with the Gencode release 28 as the reference annotations using STAR version 2.7.2b [116], followed by gene-level quantitation using htseq-count [117]. In parallel, the pre-processed reads were pseudoaligned using Kallisto version 0.43.1 [118], with 100 rounds of bootstrapping to the Gencode release 28 of the human transcriptome to which the sequence of the transfected HIV genome and the deduced HIV spliced transcripts were added. The resulting quantitation data were normalized using Sleuth. The two pipelines yielded concordant results. Pairwise differential expression tests were performed using generalized linear models as implemented in edgeR (QL) [119], and false discovery rate (FDR) values were calculated for each differential expression value.

Protein-coding genes that were expressed at a minimum abundance of 5 transcripts per million (TPM) were used for pathway analysis with fold change values as the ranking parameter while controlling false discovery rate at 0.05. Gene Set Enrichment Analysis (GSEA) package was used to identify the enriched pathway and promoter elements using mSigDB and KEGG databases [120]. Pathways that showed an FDR q-value < = 0.25 were considered significantly enriched, per the GSEA package guidelines. The number of genes contributing to the enrichment score was calculated using the leading edge output of GSEA (tag multiplied by size).

## Identification of marker genes for each study group

After filtration of the raw reads to remove low quality reads and mapping the clean reads to the human reference genome using STAR software, differential analysis was performed by edgeR package. For RNA-Seq data analysis, the bulk RNA-Seq data in a form of digital gene expression (DGE) matrix was analyzed using the Seurat package for R, v. 3.1.5 [121]. Variable

genes were identified using the *FindVariableFeatures* function. Top fifteen markers for each cluster were identified using a Wilcoxon Rank Sum test, and a heat map was generated using the *DoHeatmap* function.

## Supporting information

**S1 Fig. Toxicity testing of nuclear receptor agonists and epigenetic inhibitors.** HC69 cells were cultured in the absence or presence of the different chemicals at various concentrations for 48 hr. Cell viability of the differently treated cells was analyzed by using the eBioscience Fixable Viability Dye eFluor 450 kit (Thermo Fisher Scientific, Cat# 65–0863) and measured by flow cytometry. The average number of viable cells from each sample was calculated from triplicate assays of a single experiment. **A**, Viability of HC69 cells. Left: Cells treated with Nurr1 agonists 6-MP and AQ at 0 to 5 μM for 48 hr. Center: Cells treated with GR agonist DEXA, the RXR agonist BEXA, and the HDAC inhibitor SAHA at 0.5 to 2 μM for 48 hr. Right: Cells treated with G9a and EZH2 inhibitors UNC0638 and GSK343 at 0.625 to 2.5 μM for 48 hr. **B**, GFP expression in HC69 cells following treatment with SAHA (1 μM), UNC0638 (1.25 μM), and GSK343 (1.25 μM) for 24 hr. **C**, The GR agonist dexamethasone (DEXA, 1 μM), RXR agonist Bexarotene (BEXA, 1 μM) and 6-MP (1 μM) have additive effects on HIV silencing in HC69 cells. HC69 cells were first treated with high dose (400 pg/ml) TNF-α for 24 hr, followed by a 72 hr chase experiment during which the cells were washed with PBS and cultured in fresh media in the presence of placebo (DMSO) or the various NR agonists, alone or in combination. The expression of Nef and β-tubulin in the differently treated cells was analyzed by Western blot.
(TIF)

**S2 Fig. Regulation of HIV expression by endogenous and overexpressed Nurr1. A,** Nurr1 overexpression represses HIV in immortalized human microglial cells C20, Clone #29 cells. Western blot detection of Nef, FLAG-tagged Nurr1, and β-tubulin from clone #29 cells that stably express empty vector or 3X-FLAG-Nurr1, which were un-treated, induced with high dose TNF-α (400 pg/ml) for 24 hr, or used in a 48 hr chase after TNF-α stimulation for 24 hr. **B**, ChIP assay measurement of the levels of Nurr1 in HIV 5'LTR in the GFP+ and GFP- cells. Nurr1 was detected using a mouse monoclonal anti-Nurr1 antibody compared to control IgG. The levels of ChIP products and input DNA were quantified by qPCR by using primers (−390 F, 5'ACA CAC AAG GCT ACT TCC CTG A3', and −283 R, 5'TCT ACC TTA TCT GGC TCA ACT GGT3') specific for the Nuc-0 region. The average Nurr1 levels and error bars were calculated from triplicate technical qPCR replicates. **C,** Impact of TNF-α on endogenous expression of Nurr1. Western blot detection of Nurr1, GFP, and β-tubulin in HC69 cells at different time points post TNF-α stimulation, or in the GFP+ cells and the GFP- cells collected by cell sorting at the end of chase. **D.** Nurr1 undergoes sumoylation in microglial cells upon TNF-α stimulation. HC69 cells expressing empty vector or 3X-FLAG-Nurr1 that were stimulated with high dose (400 pg/ml) TNF-α for various time points. Total protein lysates from the differently treated cells were used for a co-immunoprecipitation (co-IP) experiment with anti-FLAG M2 magnetic beads (Millipore Sigma, Cat# M8823). The co-IP products were then subjected to Western blot analysis using the anti-FLAG antibody (**Fig 3B**) to detect 3X-FLAG-Nurr1 and a rat anti-SUMO antibody (BostonBiochem, Cat# A-714) to detect sumoylated Nurr1. The levels of β-Tubulin in the input protein samples were used as loading controls.
(TIF)

**S3 Fig. Nurr1 agonist 6-MP silences HIV by promoting recruitment of the CoREST repressor complex to HIV 5'LTR. A**, GFP expression levels measured by flow cytometry in HC69

cells expressing control shRNA or Nurr1 shRNA. Cells were either untreated or treated with 1 µM 6-MP for 48 hr, induced with high dose (400 pg/ml) TNF-α for 24 hr, and the chased in the absence or presence of 1 µM 6-MP for 72 hr after TNF-α stimulation for 24 hr. Data was calculated using three technical replicates. **B**, GFP expression levels measured by flow cytometry from HC69 cells expressing empty KO vector or G9a specific guide RNAs (G9a KO) that were treated and assayed as described in **A**. **C**, ChIP assay showing relative levels of HDAC1, CoREST, and G9a at the HIV LTR in HC69 cells that were untreated or treated with 1 µM 6-MP for 48 hr. Primers were specific for the Nuc-0 region. Data was calculated using three technical replicates.
(TIF)

**S4 Fig. Nurr1 overexpression leads to the inhibition of critical cellular proliferation pathways. A,** Patterns of differential gene expression during the chase step at 48 hr post TNF-α stimulation in vector-infected (top) and Nurr1 overexpressing (Nurr1 OE, bottom) cells. Dotted lines indicate the two-fold cut off level. **B,** Pathway analyses of Nurr1 overexpression at baseline, during TNF-α stimulation, and following the recovery period after TNF-α stimulation. The identities of specific highly enriched pathways are shown on the Y axis, and the comparisons are shown at the bottom. The color and size of circles correspond to statistical significance, as shown by FDR, and normalized enrichment values, respectively. Positive and negatively enriched pathways are shown in the left and right plot, respectively.
(TIF)

**S5 Fig. Heat maps demonstrating that Nurr1 overexpression (OE) or knock-down (KD) substantially alters the host transcriptome. A**, Heatmaps representing top 15 gene markers for each treatment group. Statistically-significant ($p < 0.001$) differentially expressed genes were determined using the Wilcoxon rank-sum test reflecting the impacts of Nurr1 OE by comparing the control cells HC69-3X-FLAG-vector (VT) with Nurr1 overexpressing cells HC69-3X-FLAG-Nurr1 (Nurr1 OE), as well as the impacts of KD by comparing the HC69-control shRNA1 and control shRNA2 (Ctl shRNA1/2) cells with HC69-Nurr1 shRNA1 and shRNA2 (Nurr1 shRNA1/2) cells, respectively. Various cell lines were cultured in the absence (untreated) or presence of high dose (400 pg/ml) TNF-α for 24 hr. In addition, cells were given 48 hr chase after stimulation with high dose (400 pg/ml) TNF-α for 24 hr and subsequent withdrawal. **B**, Heatmaps showing top 15 gene transcript markers in samples from panel **A** rearranged according to their status of treatment with TNF-α. The most enriched gene transcripts as the result of Nurr1 overexpression or KD are listed in columns to the left. The color-coded expression pattern of each gene transcript is shown in a heatmap to the right.
(TIF)

**S6 Fig. Nurr1 overexpression accelerates homeostasis of activated microglial cells by shutting down pathways involved in the maintenance of cellular activation and inflammation. A,** Identification of genes selectively altered as a result of Nurr1 overexpression (Nurr1 OE), compared to the control empty vector (Ctl VT) cells, by trajectory analysis. Genes that are unaltered (n), downregulated (d) or upregulated (u) were identified during the activation and the chase steps and were clustered into families with similar profiles. The total number of genes in each category is indicated for both the control and Nurr1-overexpressing cells. Note that the major differences in the gene expression profiles are seen in genes that are either upregulated or downregulated during the chase (highlighted by asterisks). To enable the visualization of the trajectories with low, medium and high membership, the X axis for each group is shown separately. **B,** Pathway analysis using the Hallmark gene lists of the MSigDB database was performed on non-TNF-α-responsive genes that are exclusively altered in expression

during the chase step in Nurr1 overexpressing cells, corresponding to genes which follow nnu and nnd trajectories in Nurr1 cells and an nnn trajectory in control cells (see **S7 Fig**). The identity of each pathway is shown to the left, and the direction of enrichment (+ or -) is shown at the bottom. The color and size of circles corresponded to statistical significance, as shown by FDR, and normalized enrichment values, respectively.
(TIF)

**S7 Fig. Gene trajectory analysis showing that Nurr1 overexpression mainly impacts the recovery step following TNF-α stimulation.** Trajectories of genes after stimulation with low dose (20 pg/ml) and high dose (400 pg/ml) TNF-α for 24 hr and following a 48 hr recovery (chase) after high dose TNF-α stimulation for 24 hr and subsequent withdrawal in the Nurr1 overexpression cell line HC69-3X-FLAG-Nurr1 (Nurr1 OE) were shown. Trajectories of the same genes in the control cell line HC69-3X-FLAG-vector (Ctl VT) were also shown, with a semi-transparent line connecting identical genes between the control and Nurr1-overexpressing sides of each graph. Each line represented a gene, and the Y axis values indicated the log2 expression levels. The number of genes showing each trajectory in Nurr1-overexpressing cells was shown on top. Genes that showed no change, were up regulated, and down regulated in statistically significant manner (FDR<0.05, fold change>2) were indicated with the letters n, u, and d respectively. Grouping of the different trajectories was based on gene responses during stimulation with low dose (Step 1) and high dose (Step 2) TNF-α and the recovery time after TNF-α stimulation and subsequent withdrawal (Step 3). For instance, the group of genes marked "ndu" represented genes that were not significantly changed in response to stimulation with low dose TNF-α but were down regulated with high dose TNF-α stimulation and then up regulated during the recovery (chase) period.
(TIF)

**S8 Fig. Impact of Nurr1 overexpression on cellular gene expression.** Genes that showed a different trajectory after TNF-α stimulation for 24 hr and following a 48 hr recovery period between HC69-3X-FLAG-vector (control) and HC69-3X-FLAG-Nurr1 cells were identified and groups containing over 100 genes were graphed. Each line represented a gene and a semi-transparent line connected identical genes between control and Nurr1-overexpressing sides of each graph. The Y axis indicated the expression level of each gene throughout the trajectory. Grouping of genes with no statistically significant changes in expression (n), up regulated (d), or down regulated (d) in the three steps of reactivation and chase experiment.
(TIF)

**S9 Fig. Nurr1 overexpression (OE) substantially alters host transcriptome.** Heatmaps showing the genes involved in the top negatively enriched pathways in Nurr-1 overexpressing cells. **A**, MYC Targets. **B**, E2F Targets. **C**, G2M Checkpoint. **D**, IFN Response. The values shown in the heatmap correspond to the level of differential expression between Nurr1 overexpressing cells (marked as "Nurr1") versus vector-infected control cells (marked as "Vector") during the chase step. The identities of the plotted pathways and genes involved in the pathways are shown on the top and to the right, respectively.
(TIF)

**S10 Fig. Nurr1-specific gene expression during the chase step leads to strong downregulated of genes involved in cell cycle.** Genes that exclusively change in expression during the chase step only in Nurr1 overexpressing (Nurr1 OE) cells (see **S6 Fig**) were superimposed on the KEGG cell cycle graph [120]. The color bar on the top right indicates the level of differential expression for each gene in Nurr1 cells during the chase step.
(TIF)

**S11 Fig. TNF-α stimulation leads to strong induction of NF-κB-responsive genes along with targets of multiple inflammatory cytokines.** The most enriched transcription factor binding motifs in proximity of the promoters of differentially expressed genes are shown. The size of the circles indicates the level of enrichment, while the color intensity reflects the statistical significance as shown by FDR. Positively- and negatively-enriched motifs are shown after each treatment (shown at the bottom) in the left and right panel, respectively. The identity of each motif, as annotated in the C3 lists of the MSIGDB database, is shown to the left.
(TIF)

**S12 Fig. Analysis of genes carrying Nurr1 binding sites.** Genes with NURR1 binding sites within 2000 nucleotides of their locus were identified using the NURR1 ChIP-seq studies reported by the ENCODE project. **A**. Pathway analysis of genes that were negatively enriched or positively enriched in Nurr-1 overexpressing cells during the chase step. **B**. Volcano plots showing the distribution of Nurr-1 dependent differentially expressed genes carrying one or more Nurr1 sites. Top: Untreated cells. Middle: Cells activated with 400 pg/ml TNF-α for 24 hr. Bottom: Cells chased after activation for 72 hrs. HIV is highlighted in purple. Note that the majority of Nurr1 binding site-containing genes were upregulated as a result of Nurr1 overexpression. However a fraction of genes, including HIV were downregulated under these conditions.
(TIF)

## Acknowledgments

We thank Meenakhi Shukla for technical assistance for production of HIV-1 reporter virus. This work made use of the High Performance Computing Resource for Advanced Research Computing and flow cytometry and virology cores of the Center for AIDS Research (CFAR) at Case Western Reserve University.

## Author Contributions

**Conceptualization:** Fengchun Ye, David Alvarez-Carbonell, Jonathan Karn.

**Data curation:** Fengchun Ye, David Alvarez-Carbonell, Kien Nguyen, Konstantin Leskov, Saba Valadkhan.

**Funding acquisition:** Jonathan Karn.

**Investigation:** Fengchun Ye, Kien Nguyen, Yoelvis Garcia-Mesa, Sheetal Sreeram.

**Methodology:** Fengchun Ye, Kien Nguyen.

**Project administration:** Jonathan Karn.

**Resources:** Jonathan Karn.

**Supervision:** Jonathan Karn.

**Writing – original draft:** Fengchun Ye, David Alvarez-Carbonell, Jonathan Karn.

**Writing – review & editing:** Fengchun Ye, David Alvarez-Carbonell, Kien Nguyen, Konstantin Leskov, Yoelvis Garcia-Mesa, Sheetal Sreeram, Saba Valadkhan, Jonathan Karn.

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
