## [Decision Letter · Decision Letter 0]

10 Jan 2022

Dear Dr Fengchun Ye,

Thank you very much for submitting your manuscript "The Nerve Growth Factor IB-like Receptor Nurr1 (NR4A2) recruits CoREST transcription repressor complexes to silence HIV following proviral reactivation in microglial cells." for consideration at PLOS Pathogens. As with all papers reviewed by the journal, your manuscript was reviewed by members of the editorial board and by several independent reviewers. We and the reviewers appreciated the relevance of the study with findings that could be important for HIV latency in the CNS. 

In light of the reviews (below this email), we would like to invite the resubmission of a significantly-revised version that takes into account the reviewers' comments. Indeed, because the manuscript mainly extends the list of nuclear receptors (glucocorticoid and estrogen receptors) that inhibit HIV transcription, we feel it would be important to get some complements on the mechanism of action of Nurr1, and notably to know whether Nurr1 binds DNA. One major concern to address  is about ligand-mediated repression, whether it would involve the same mechanism than silencing following TNFalpha (reviewer 1). As a minor concern, we would like to see Nurr1 expression in the GFP- and GFP+ populations at the end of the chase experiment (lane 159). We would expect to see better expression of Nurr1 in the GFP- population (reviewer 2). Doing the 6-MP suppression experiment using Nurr1 KO cells would also be appreciated as a control. Finally, the authors should comment on the use of a clonal cell line and provide a detailed response to all the points raised by the reviewers.

We cannot make any decision about publication until we have seen the revised manuscript and your response to the reviewers' comments. Your revised manuscript is also likely to be sent to reviewers for further evaluation.

Sincerely,

Florence Margottin-Goguet

Associate Editor

PLOS Pathogens

Alexandra Trkola

Section Editor

PLOS Pathogens

Kasturi Haldar

Editor-in-Chief

PLOS Pathogens

orcid.org/0000-0001-5065-158X

Michael Malim

Editor-in-Chief

PLOS Pathogens

orcid.org/0000-0002-7699-2064

We and the reviewers appreciated the relevance of the study with important findings for HIV latency in the CNS. Therefore, we would like to invite the resubmission of a significantly-revised version that takes into account the reviewers' comments. Indeed, because the manuscript mainly extends the list of nuclear receptors (glucocorticoid and estrogen receptors) that inhibit HIV transcription, we feel it would be important to get some complements on the mechanism of action of Nurr1, and notably to know whether Nurr1 binds DNA. One major concern is about ligand-mediated repression, whether it would involve the same mechanism than silencing following TNFalpha (reviewer 1). As a minor concern, we would like to see Nurr1 expression in the GFP- and GFP+ populations at the end of the chase experiment (lane 159). We would expect to see better expression of Nurr1 in the GFP- population (reviewer 2). Doing the 6-MP suppression experiment using Nurr1 KO cells would also be appreciated as a control. Finally, the authors should comment on the use of a clonal cell line and provide a detailed response to all the points raised by the reviewers.

Reviewer's Responses to Questions

**Part I - Summary**

Reviewer #1: The paper by Ye et al explores the ability of the nuclear receptor, Nurr1, to repress HIV-1 transcription in the context of microglia. The relevance of these studies is it addresses mechanisms that may contribute to the establishment and maintenance of HIV latency in microglia, a reservoir in the brain, as well as potential pathways that may contribute to CNS inflammatory comorbidities. Evidence presented that supports the conclusion that Nurr1 represses HIV transcription includes Nurr1 agonist repressing transcription in immortalized human microglia cells and iPSCs (the iPSC derived microglia maybe the strongest and most provocative data), and overexpression and knockdown experiments. Furthermore, repression was associated with recruitment of a CoREST associated repressor complex. Most experiments were performed with TNFa and assessing how Nurr1 influenced the ability of this cytokine to induce HIV-1 transcription in microglia. These findings provide an interesting foundation for understanding the control of HIV in microglia by Nurr1 but it also rises many questions. For example, is ligand dependent repression similar to the silencing seen in response to TNFa? How is Nurr1 regulated? Does Nurr1 bind DNA or mediate its activity through a trans-repression mechanism? Furthermore, although interesting, the transcriptome data seems to be a tangent and distracts from the focus on Nurr1 transcriptional mechanisms. Some specific comments are below.

Reviewer #2: In the manuscript “The Nerve Growth Factor IB-like Receptor Nurr1 (NR4A2) recruits CoREST transcription repressor complexes to silence HIV following proviral reactivation in microglial cells”, Ye et al and colleagues report on their investigation of the Nerve Growth Factor IB-like nuclear receptor Nurr1 (NR4A2 and its role in restricting HIV transcription in microglia cells. The authors’ major finding is that Nurr1 plays an important role in shutting down HIV transcription in microglia cells via the recruitment of the CoREST/HDAC1/G9a/EZH2 transcription repressor complex to the HIV promoter. Using transcriptomic analyses, Ye et al also report that Nurr 1 downregulates several cellular genes associated with inflammation, cell cycle and metabolism to promote HIV silencing and microglia homeostasis.

Major Comments

The authors have made some important observations about the role of Nurr1 in HIV transcription in microglia cells. The manuscript is clearly written and provides partial explanation for the silencing of HIV transcription in microglia following spontaneous reactivation that the authors observed in their previous work. However, there are some limitations to this study. Namely, all the mechanistic work was performed using a transformed microglia cell line. Understandably, accessing brain microglia from ART-treated individuals is impossible, but newly established resources such as the Last Gift Study ( Rawlings SA et al, PMID: 34257439) could serve as a post-mortem source of well-preserved microglia cells from ART-infected people to at the very least confirm some of the observations made with agonists and epigenetic drugs. The iPSC derived human microglial cells infected in vitro with a reporter virus while provide some information, still lack the nuances of microglial cells isolated from HIV-infected individuals. Additional points that should be addressed are listed below.

1. The HC69 represents a clonal population of cells with the same integration site. Would the authors conclusion hold true in primary microglia cells isolated from ART-treated individuals where there is likely to be multiple integration site in the microglial cell population?

2. The authors concentrated on Nurr1, however, the western blot analysis in figure 3D would also suggest at least a partial role for Nur77 and Nor1 in HIV silencing.

3. Figure 8A, As these epigenetic drugs are reported to induce HIV expression at least in T cells, an important control to include is the effect of these compounds on cells that have not been treated with TNF-α. Are they simply reactivating silent HIV genomes?

Reviewer #3: Ye etal describe the activity of Nurr1 in suppressing HIV following reactivation in microglial cells. The authors demonstrate this using Nurr1 overexpression, knockout cells, and treatment with the Nurr1 agonist 6-MP. They further demonstrate the mechanism of action is through the CoREST repressor complex. Given most of these experiments are performed in a latent cell line model, the authors also crucially demonstrate HIV suppression in an infection model using iPSC-microglial cells. The authors data also hints at an anti-inflammatory activity of Nurr1 following HIV reactivation which has the potential to be very significant given the role of HIV activity in HAND. Overall, this is a very well written paper (albeit dense) with important findings for HIV latency in the CNS, HAND and the CNS reservoir and will be of interest to those in the field.

**Part II – Major Issues: Key Experiments Required for Acceptance**

Reviewer #1: 1) Initial experiments examined repression of spontaneous activation with Nurr1 agonist. It was not clear as to why ligand mediated repression was not explored in greater detail. It also raises questions as to whether the mechanisms of ligand mediated repression and silencing following TNFa are similar.

2) Does overexpression or knocking down Nurr1 alter the kinetics of activation and repression (spontaneous activation or silencing following TNFa treatment)?

3) Overexpressed Nurr1 was demonstrated to Co-IP with the CoREST repressor complex. Was overexpression necessary? Could endogenous Nurr1 pulldown CoREST?

4) The ChIP of CoREST complexes is interesting. The result suggests follow-up experiments to determine if this correlates with repressive histone post-translational marks. Also, it appears CoREST, G9a and EZH2 are associated with and represses promoters HIV in unstimulated cells (Fig 8) begging the question if these complexes display regulated binding during spontaneous activation?

5) Nuclear receptors have been shown to activate and repress genes by ligand dependent DNA binding or through trans mechanisms independent of DNA binding. There are no experiments that provide insight into Nurr1 mechanism of action.

Reviewer #2: 4. Line 161, the authors mentioned that the substantial decrease in GFP+ observed in the chase experiments is unlikely due to GR (glucocorticoid receptor )-mediated HIV silencing as the cells were not cultured in the presence of dexamethasone, a ligand of GR. What were the culture conditions for the cells? This is missing in the methods section. The presence of fetal bovine serum (FBS) could be a potential source of cortisol (GR ligand) to which the cells are exposed to.

5. What is the effect of toxicity of the agonists and epigenetic drugs used on some of the observations? The authors should provide toxicity data on the microglia cells.

6. Line 383 and figure 7, there is really no strong evidence to suggest that G9a and HDAC1 decreased at the promoter at 4hr post-TNF-α treatment. In fact, this is clearly not the case in the qPCR results. The minor differences in the peaks in the ChiP seq experiment could be due to experimental fluctuations. Did the authors use a normalization approach, i.e spike in a known quantity of a certain DNA to standardized between samples (e.g., ChIP seq spike in normalization by Active Motif). Also, are the scales the same for the read counts (since they are not shown)? How many times were these experiments performed?

7. In fact in many of the figures , it is not cleared how many times the experiments were repeated (e.g. western blot analyses data). The number of independent experiments performed to demonstrate reproducibility should be clearly stated for all the figures (not just the replicates). Also, the type of statistical tests performed should be indicated in both the figure legends and a statistical analyses plan should be included in the methods.

Reviewer #3: If indeed Nurr1 is silencing HIV transcription, a useful confirmatory experiment would be to sort the GFP+ and GFP- cells at the end of your chase experiment (61% GFP +) and determine Nurr1 expression (RNA, protein) in these two populations. One would expect higher levels of Nurr1 in GFP- cells. This experiment could be performed using the iPSC- infection model.

It would be useful to confirm that 6-MP is exerting its effect through nurr1 by performing the 6-MP suppression experiment using the Nurr1 knockout cells. This is especially important given the confirmatory experiments in iPSC were done with 6-MP.

**Part III – Minor Issues: Editorial and Data Presentation Modifications**

Reviewer #1: 1) Figure 1 may not be necessary or use as supplemental information. They have described the cell line in previous experiments and used it to examine glucocorticoid receptors and Dex mediated repression of HIV in previous publications.

2) The transcriptomic experiments are interesting and indicate a role for Nurr1 microglia but are a tangent to understanding how Nurr1 controls HIV expression. It would also be interesting to include Nurr1 ligand and compare to cells overexpressing Nurr1.

Reviewer #2: Minor Comments

1. Line 69 appears to be incomplete: “However, the early studies neglected both the side effects of anti-HIV drugs on neuronal damage which could mask the benefits of reduced HIV expression by cART and the impact of HIV latency on the development of HAND”. Both the side effects of anti HIV drugs and what else?

2. As this is still controversial and under intense investigation, can the authors confirm their statement that microglia are “long-lived” reservoirs in ART-suppressed individuals (line 76). The reference provided (Ref 20) does not provide concrete evidence.

3. In the methods section, instead of randomly listing the antibodies used in the study, to help guide the readers, the antibodies should be listed in accordance with the assay it was used in (e.g. ChiP, immunoblotting, etc.). Are the ChiP antibodies validated for the purpose of chromatin immunoprecipitation for example.

Reviewer #3: Line 64-65: ART lowers HIV RNA in the brain – the first article is an SIV paper, the second uses samples from PLHIV however the majority of them were not virally suppressed, can we find some better references?

Line 76-77: given the persistence of HIV in CNS remains a controversial topic, perhaps this assertations could be backed up be primary articles rather than a review article?

Figure 1 spontaneous silencing, could this not just be outgrowth of GFP- cells?

Figure 2A: toxicity of 6-MP in cells?

Figure 2B: Intense Nur77 band with 5uM 6-MP, but this doesn’t titrate. Is this an error? Can authors address.

Figure 2C: I don’t think this data is really useful or necessary. Also I don’t think you can use nef WB to look at this, should be GFP. If authors want to assess 6-MP synergy then they should do proper synergy experiments and use the GFP reporter rather than nef WB

Figure 3A: Only Nurr1 shown, perhaps this is better shown with a bar graph of total reads (and include all three genes)?

Figure 3B: Expression of Nurr1 and Nor1 much less than previously seen (figure 2B), even though tubulin bands have higher intensity?

Figure 4A: Enumeration of the magnitude of Nurr1 knockdown is required. The way the data is currently presented does not give this sense.

Figure 5: At what stage during the chase is the RNAseq performed, this is not made clear?

Figure 5: How many replicates were these RNAseq data derived from?

Figure 7C, D: Do data points represent mean and SD of technical replicates from one experiment? Or means from multiple independent experiments?

Line 403-404: References for the activity of the described drugs would be useful for reader confirmation

Figure 9: enumeration of cell viability with these different combinations of HIV infection and drug treatment are important.

Figure 9 figure legend C: assume you mean three independent experiments and not technical replicates?

Figure 9D: The unspliced RNA data is not presented clearly. Assume fold change is calculated from the no treatment wells, yet it is not clear this is 1 given the strange choice in axis (0.9, 1.2?). What are there * above untreated? If this is a comparison to uninfected then this needs to be made clear in the figure legend.

Discussion point: Is Nurr1 expressed/active in macrophage cells? Could a similar mechanism be at work in infected macrophages in the perivascular space?

Editing

Line 169: double period

Line 209: insert ‘and’ between studies and western

PLOS authors have the option to publish the peer review history of their article (what does this mean?). If published, this will include your full peer review and any attached files.

Reviewer #1: No

Reviewer #2: No

Reviewer #3: No
---

## [Decision Letter · Decision Letter 1]

15 Jun 2022

Dear Dr Fengchun Ye,

We are pleased to inform you that your manuscript 'Recruitment of the CoREST Transcription Repressor Complexes by Nerve Growth Factor IB-like Receptor (Nurr1/NR4A2) Mediates Silencing of HIV in Microglial Cells' has been provisionally accepted for publication in PLOS Pathogens.

Best regards,

Florence Margottin-Goguet

Associate Editor

PLOS Pathogens

Alexandra Trkola

Section Editor

PLOS Pathogens

Kasturi Haldar

Editor-in-Chief

PLOS Pathogens

orcid.org/0000-0001-5065-158X

Michael Malim

Editor-in-Chief

PLOS Pathogens

orcid.org/0000-0002-7699-2064

Reviewer Comments (if any, and for reference):

Reviewer's Responses to Questions

**Part I - Summary**

Reviewer #1: The authors have added significant new data that provide additional mechanistic insights in the revised manuscript. All concerns of this reviewer have been appropriately addressed and/or thoughtfully considered.

Reviewer #2: No additional comments

Reviewer #3: Ye etal have done a commendable job addressing comments from each reviewer. The new data added greatly enhances the manuscript and goes some way to addressing the mechanism behind Nurr1 repression of HIV reactivation. I am satisfied that the authors have addressed my comments and have no further questions.

**Part II – Major Issues: Key Experiments Required for Acceptance**

Reviewer #1: There are no major issues with the revised manuscript.

Reviewer #2: (No Response)

Reviewer #3: N/A

**Part III – Minor Issues: Editorial and Data Presentation Modifications**

Reviewer #1: All minor issues have been addressed.

Reviewer #2: (No Response)

Reviewer #3: N/A

PLOS authors have the option to publish the peer review history of their article (what does this mean?). If published, this will include your full peer review and any attached files.

Reviewer #1: No

Reviewer #2: No

Reviewer #3: **Yes: **Michael Roche

---

## [Editor Report · Acceptance letter]

4 Jul 2022

Dear Dr. Ye,

We are delighted to inform you that your manuscript, "Recruitment of the CoREST Transcription Repressor Complexes by Nerve Growth Factor IB-like Receptor (Nurr1/NR4A2) Mediates Silencing of HIV in Microglial Cells," has been formally accepted for publication in PLOS Pathogens.

Best regards,

Kasturi Haldar

Editor-in-Chief

PLOS Pathogens

orcid.org/0000-0001-5065-158X

Michael Malim

Editor-in-Chief

PLOS Pathogens

orcid.org/0000-0002-7699-2064